# Earth System Music: the methodology and reach of music generated from the United Kingdom Earth System Model (UKESM1)

Lee de Mora[1], Alistair A. Sellar[2], Andrew Yool[3], Julien Palmieri[3], Robin S. Smith[4,5], Till Kuhlbrodt[4], Robert J. Parker[6,7], Jeremy Walton[2], Jeremy C. Blackford[1], and Colin G. Jones[8]

[1]Plymouth Marine Laboratory, Plymouth, UK
[2]Met Office Hadley Centre, Exeter, UK
[3]National Oceanography Centre, Southampton
[4]National Centre for Atmospheric Science, Department of Meteorology, University of Reading, Reading, UK.
[5]University of Reading, Reading, UK
[6]National Centre for Earth Observation, Leicester, UK
[7]University of Leicester, Leicester, UK
[8]National Centre for Atmospheric Science, School of Earth and Environment, University of Leeds, Leeds, UK

**Correspondence:** Lee de Mora (ledm@pml.ac.uk)

**Abstract.**

Scientific data is almost always represented graphically in figures or in videos. With the ever-growing interest from the general public towards understanding climate sciences, it is becoming increasingly important that scientists present this information in ways that are both accessible and engaging to non-experts.

In this pilot study, we use time series data from the first United Kingdom Earth System model (UKESM1) to create six procedurally generated musical pieces. Each of these pieces presents a unique aspect of the ocean component of the UKESM1, either in terms of a scientific principle or a practical aspect of modelling. In addition, each piece is arranged using a different musical progression, style and tempo.

These pieces were created in the Musical Instrument Digital Interface (MIDI) format and then performed by a digital piano synthesizer. An associated video showing the time development of the data in time with the music was also created. The music and video were published on the lead author's YouTube channel. A brief description of the methodology was also posted alongside the video. To begin the dissemination of these works, a link to each piece was published using the lead authors personal and professional social media accounts.

The reach of these works was analysed using YouTube's channel monitoring toolkit for content creators, YouTube studio. In the first ninety days after the first video was published, the six pieces reached at least 251 unique viewers, and have 553 total views. We found that most of the views occurred in the fourteen days immediately after each video was published. We also discuss the limitations of this pilot study, and describe several approaches to extend and expand upon this work.

## 1 Introduction

The use of non-speech audio to convey information is known as sonification. One of the earliest and perhaps the most well known applications of sonification in science is the Geiger counter; a device which produces a distinctive clicking sound when it interacts with ionising radiation (Rutherford and Royds, 1908). Beyond the Geiger counter, sonification is also widely used in monitoring instrumentation. Sonification is appropriate when the information being displayed changes in time, includes warnings, or calls for immediate action. Sonification instrumentation is used in environments where the operator is unable to use a visual display, for instance if the visual system is busy with another task, overtaxed, or when factors such as smoke, light, or line of sight impact the operators visual system (Walker and Nees, 2011). Sonification also allows several metrics to be displayed simultaneously using variations in pitch, timbre, volume and period (Pollack and Ficks, 1954; Flowers, 2005). For these reasons, sonification is widely used in medicine for monitoring crucial metrics of patient health (Craven and Mcindoe, 1999; Morris and Mohacsi, 2005; Sanderson et al., 2009).

Outside of sonification for monitoring purposes, sonification of data can also be used to produce music. There have been several examples of sonification of climate system data. *Climate symphony* by Disobedient films, (Borromeo et al., 2016) is a musical composition performed by strings and piano using observational data from sea ice indices, surface temperature and carbon dioxide concentration. Daniel Crawford's *Planetary Bands, Warming World*, (Crawford, 2013) is a string quartet which uses observational data from the Northern Hemisphere temperatures. In this piece, each of the four stringed parts represents a different latitude band of the Northern Hemisphere temperature over the time range 1880-2012. Similarly, the climate music project, https://climatemusic.org/, is a project which makes original music inspired by climate science. They have produced three pieces which cover a wide range of climatological and demographic data and both observational and simulated data. However, pieces like (Borromeo et al., 2016) and (Crawford, 2013) often use similar observational temperature and carbon dioxide datasets. Both of these datasets only have monthly data and approximately one century of data or less available. In addition, both temperature and carbon dioxide have risen since the start of the observational record. This means that these musical pieces tend to have similar structures and sounds. The pieces start slowly, quietly and low pitched at the start of the dataset, then slowly increase, building up to a high pitch conclusion at the present day. It should be noted that all the pieces list here are also accompanied by a video which can explain the methodology behind the creation of the music, shows the performance by the artists, or shows the data development while the music is played.

An alternative strategy was deployed in the Sounding Coastal Change project (Revill, 2018). In that work, sound works, music recordings, photography and film produced through the project were geotagged and shared on to a sound map. This created both a record of the changing social and environmental soundscape of North Norfolk. They used these sounds to create music and explore the ways in which the coast was changing and how people's lives were changing with it.

In addition to its practical applications, sonification is a unique field where scientific and artistic purposes may coexist (Tsuchiya et al., 2015). This is especially true when in addition to being converted into sound, the data is also converted into music. This branch of sonification is called musification. Note that the philosophical distinction between sound and music is beyond the scope of this work. Through the choice of musical scales and chords, tempo, timbre and volume dynamics, the composer adds emotive meaning to the piece. As such, unlike sonification, musification should be treated as a potentially biased-interpretation of the underlying data. It can not be a true objective representation of the data. Furthermore, even though the composer may have made musical and artistic decisions to link the behaviour of the data with an emotive state, it may not necessarily be interpreted in the same way by the listener.

With the ever-growing interest from the general public towards understanding climate science, it is becoming increasingly important that we present our model results and methods in ways that are accessible and engaging to non-experts. It is also becomingly increasingly easier for scientists to use tools such as social media to engage with non-expert audiences and the wider public.

In this work, six musical pieces were procedurally generated using output from a climate model; specifically, the first version of the United Kingdom Earth System Model (UKESM1) (Sellar et al., 2019). By using simulated data instead of observational data, we can reach time periods beyond the recent past such as the pre-industrial period before 1850 and multiple projections of possible future climates. Similarly, model data allows access to regions and measurements far beyond what can be found in the observational record. The UKESM1 is a current generation computational simulation of the Earth's climate and has been deployed to understand the historical behaviour of the climate system as well as make projections of the climate in the future. The UKESM1 is described in more detail in sec. 2. The methodology used to produce the pieces and a brief summary of each piece is shown in sec. 3. The aims of the project are outlined below in sect. 3.1.

Each of the six musical pieces was produced alongside a video showing the time series data developing concurrently with the music. These videos were published on the YouTube video hosting service and shared via the author's personal and professional social media network. In addition to hosting the video, YouTube also provides YouTube studio, a powerful channel monitoring toolkit for content creators (Google, 2019). This toolkit allows content creators to monitor the reach, engagement and audience demographics (age, gender, country of origin) for their channel as a whole, as well as for individual videos. This work includes a study on the reach of these pieces using these tools. Section 3.2 contains a brief summary of the methods used to measure the reach. The results of this analysis are shown in sec. 4, and a discussion is in sec. 5. It should be noted that this work is an early pilot study and the limitations of this approach are outlined in sect. 6.

## 2 UKESM1

The UKESM1 is a computational simulation of the Earth System produced by a collaboration between the Hadley Centre Met Office from the United Kingdom and the Natural Environment Research Council (NERC) (Sellar et al., 2019). The UKESM1 represents a major advance in Earth System modelling, including a new atmospheric circulation model with a well-resolved stratosphere; terrestrial biogeochemistry with coupled carbon and nitrogen cycles and enhanced land management;

troposphere-stratospheric chemistry allowing the simulation of radiative forcing from ozone, methane and nitrous oxide; a fully featured aerosol model; and an ocean biogeochemistry model with two-way coupling to the carbon cycle and atmospheric aerosols. The complexity of coupling between the ocean, land and atmosphere physical climate and biogeochemical cycles in UKESM1 is unprecedented for an Earth System model.

In this work, we have exclusively used data from the ocean component of the UKESM1. The UKESM1's ocean is subdivided into three component models: the Nucleus for European Modelling of the Ocean (NEMO) simulates the ocean circulation and thermodynamics (Storkey et al., 2018), the Model of Ecosystem Dynamics, nutrient Utilisation, Sequestration and Acidification (MEDUSA) is the sub model of the marine biogeochemistry (Yool et al., 2013) and CICE simulates the growth, melt and movement of sea ice (Ridley et al., 2018).

The UKESM1 is being used in the UK's contribution to the sixth international coupled model intercomparison project (CMIP6) (Eyring et al., 2016). The UKESM1 simulations that were submitted to the CMIP6 were used to generate the musical pieces. These simulations include the pre-industrial control (piControl), several historical simulations and many projections of future climate scenarios. The CMIP6 experiments that were used in these works are listed in tab. 1.

This is not the first time that the UKESM1 has been used to inspire creative projects. In 2017, the UKESM1 participated in a science and poetry project where a scientist and a writer were paired together to produce poetry. Ben Smith was paired with L. de Mora and produced several poems inspired by the United Kingdom Earth System Model (Smith, 2018).

## 3    Methods

In this section, we describe the method used to produce the music and the videos. Figure 1 illustrates this process. The initial data is UKESM1 model output files, downloaded directly from the United Kingdom's Met Office's data storage system, MASS. These native-format UKESM1 data will not be available outside the UKESM collaboration, but selected model variables have been transformed into a standard format and made available on the Earth System Grid Federation via, for example, https://esgf-index1.ceda.ac.uk/search/cmip6-ceda/.

The time series data is calculated from the UKESM1 data by the BGC-val model evaluation suite (de Mora et al., 2018). BGC-val is a software toolkit that was deployed to evaluate the development and performance of the ocean component of the UKESM1. In all six pieces, we use annual average data as the time series data. The datasets that were used in this work are listed in tab. 1.

Each time series dataset is used to create an individual Musical Instrument Digital Interface (MIDI) track composed of a series of MIDI notes. The MIDI protocol is a standardised digital way to convey musical performance information. It can be thought of as instructions that tell a music synthesizer how to perform a piece of music (The MIDI Manufacturers Association, 1996). All six pieces shown here are saved as a single MIDI file, which contains one or many MIDI tracks played simultaneously. Each MIDI track is composed of a series of MIDI notes.

Each MIDI note is assigned four parameters. The first two parameters are timing: when the note occurs in the song, and duration: the length of time that the note is held. The timing is the number of beats between this note and the beginning of the

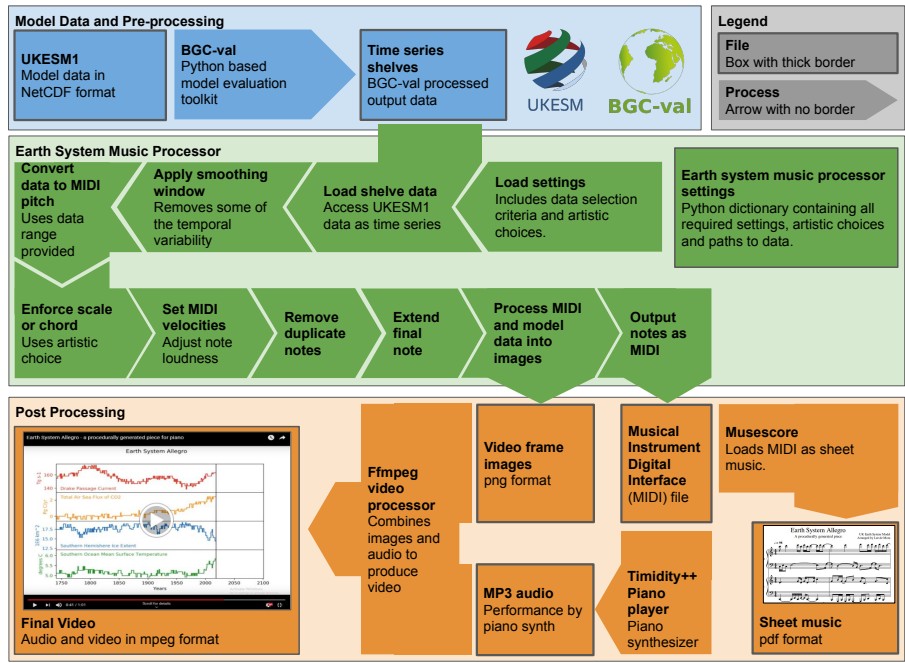

**Figure 1.** The computational process used to convert UKESM1 data into a musical piece and associated video. The boxes with a dark border represent files and datasets, and the arrows and chevrons represent processes. The blue areas are UKESM1 data and the pre-processes stages, the green areas show the data and processing stages needed to convert model data into music in the MIDI format, and the orange area shows the post processes stages which convert images and MIDI into sheet music and videos.

song. The duration is positive rational number representing the number of beats for the note to be held. A unity duration is equivalent to a crotchet (quarter note), a duration of two is a minim (half note), a duration value of a half is a quaver (eighth note).

The third MIDI note parameter is the pitch, which in MIDI must be an integer between 1 and 127, where 1 is a very low pitch and 127 is a very high pitch. These integer values represent the chromatic scale and middle-C is set to a value of 60. The pitch of the MIDI notes must be an integer, as there is no capability in MIDI for notes to sit between values on the chromatic scale. Musically, this can be explained that there are not notes in-between the notes on a keyboard in MIDI. The total range of available pitches covers ten and a half octaves, however we found that pitches below 30 or above 110 started to become unpleasant when performed by TiMidity; other MIDI pianos may have more success. Also note that MIDI's 127 note system extends beyond the standard piano keyboard which only covers the range 21-108 of the MIDI pitch system. MIDI uses the twelve tone equal temperament tuning system - while this is not the only tuning system, it is the most widely used in Western music.

The fourth MIDI note parameter is the velocity; this indicates the speed with which the key would be struck on a piano and is the relative loudness of the note. In practical terms, velocity is an integer ranged between 1 and 127 where 1 is very quiet

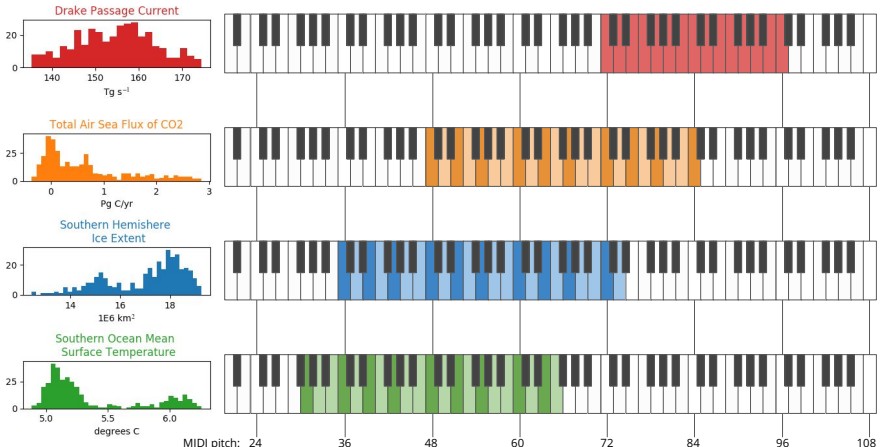

**Figure 2.** The musical range of each of the datasets used in the *Earth System Allegro*. The four histograms on the left hand side show the distributions of data used in the piece, and the right hand side shows a standard piano keyboard which the musical range available to each dataset. In this piece, the Drake Passage current, shown in red, is free to vary within a two octave range of the C major scale. The other three datasets have their own ranges, but are limited to the notes in the chord progression C major, G major, A minor F major. The dark coloured keys are the notes in C major chord, but the lighter coloured keys show the other notes which are available for the other chords in the progression. Note that both the C major scale and chord do not include any of the ebony keys on a piano, but these notes could be used if they were within the available range and appeared in the chord progression used.

and 127 is very loud. The overall tempo of the piece is assigned as a global parameter of the MIDI file in units of the number of beats per minute.

Each model timeseries dataset is converted into a series of consecutive MIDI notes, which form a track. For instance, the Sea Surface Temperature (SST) time series could be converted into a series of MIDI notes in the upper range of the keyboard,
forming a track. For each track, the time series data is converted into musical notes such that the lowest value in the dataset is represented by the lowest note pitch available, and the highest value of the dataset is represented by the highest pitch note available. The notes in between are assigned proportionally by their data value between the lowest and highest pitched notes. The lowest and highest notes available for each track are pre-defined in the piece's settings and they are considered an artistic decision. Each track is given its own customised pitch range, so that the tracks may be lower pitch, higher pitch or have
overlapping pitch ranges relative to other tracks in the piece. The ranges of notes available for the piece *Earth System Allegro* is shown in fig. 2. In this figure, the four histograms on the left hand side show the distributions of data used in the piece, and the right hand side shows a standard piano keyboard which the musical range available to each dataset. For instance, the Drake Passage Current ranges between 135 and 175 Tg s$^{-1}$ in these simulations and we selected a range between MIDI pitches 72 and 96. This means that the lowest Drake Passage current values (135 Tg s$^{-1}$) would be represented in MIDI with a pitch of 72
and the highest Drake Passage current values (175 Tg s$^{-1}$) would be assigned a MIDI pitch of 96, which is two octaves higher.

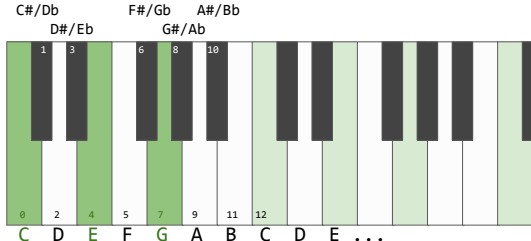

**Figure 3.** A depiction of a standard piano keyboard, showing the names of the notes, the number of these notes in MIDI format. The C major chord is highlighted in green, and the zeroth octave is shown in a darker green than the subsequent octaves.

These note pitches are then binned into a scale or a chord. The choice of chord or scale depends on the artistic decisions made by the composer. For instance, the C major chord is composed of the notes C, E and G, which are the $0^{th}$, $4^{th}$ and $7^{th}$ notes respectively in the 12 note chromatic scale starting from C at zero. Figure 3 shows a representation of these notes on a standard piano keyboard. The C major in the zeroth octave is composed of the following set of MIDI pitch integers:

$$C_{maj_0} = \{0, 4, 7\} \tag{1}$$

In the twelve tone equal temperament tuning system, the twelve named notes are repeated and each distance of 12 notes represents an octave. As shown in fig. 3, a chord may also include notes from subsequent octaves. In this figure, the C major chord is highlighted in green, and the zeroth octave is shown in a darker green than the subsequent octaves. As such, the C major chord can be formed from any of the following set of MIDI pitches:

$$C_{maj_{0,1,2...}} = \{0, 4, 7, 12, 16, 19, 24, 28, 31 \ldots 127\} \tag{2}$$

It then follows that the notes of the C major chord are values between 0 an 127 where the condition is true:

$$p \in C_{maj_{0,1,2...}}$$

This can be can be written more simply as:

$$p \,\% \, 12 \in C_{maj_0}$$

where $p$ represents the pitch value: an integer between the minimum and maximum pitches provided in the settings, and the percent sign (%) represents the remainder operator.

The zeroth octave values for other chords and scales with the same root note can be calculated from their chromatic relation with the root note. For instance:

$$C_{min_0} = \{0, 3, 7\}$$
$$C^7_{maj_0} = \{0, 4, 7, 11\}$$
$$C^7_{min_0} = \{0, 3, 7, 10\}$$

...

Note that the derivation of these chords and their nomenclature is beyond the scope of this work. For more information on music theory, please consult an introductory guide to music theory such as Schroeder (2002) or Clendinning and Marvin (2016).

The zeroth octave values for other keys can be included by appending the root note of the scale ($C : 0$, $C\#/Db : 1$, $D : 2$, $D\#/Eb : 3$ and so on) to the relationships in the key of C above. For instance,

$$C_{maj_0} = \{0, 4, 7\}$$
$$C\#_{maj_0} = \{0, 4, 7\} + 1 = \{1, 5, 8\}$$
$$D_{maj_0} = \{0, 4, 7\} + 2 = \{2, 6, 9\}$$
$$D\#_{maj_0} = \{0, 4, 7\} + 3 = \{3, 7, 10\}$$

...

Using these methods, we can combinatorially create a list of all the MIDI pitches in the zeroth octave for all 12 keys for most standard musical chords. From this list, we can convert model data into nearly any choice of chord or scale.

The conversion from model data to musical pitch is performed using the following method. First, the data is translated into the pitch scale, but kept as a rational number between the minimum and maximum pitch range assigned by the composer for this dataset. As an example, in the piece, *Earth System Allegro*, the Drake Passage current was assigned a pitch range between 72 and 96, as shown in fig.2. Once the set of possible integer pitches for a given chord or scale has been produced using the methods described above, the in-scale MIDI pitch with this smallest distance to this rational number pitch is used. As mentioned earlier, the pitch of the MIDI notes must be an integer, as there is no capability in MIDI for notes to sit between values on the chromatic scale. The choice of scale is provided in the piece's settings and is an artistic choice made by the composer. Furthermore, instead of using a single chord or scale for a piece, it is also possible to use a repeating pattern of chords or a chord progression. The choice of chords, and the order of chords are different for each piece. In addition, the number of beats between chord changes, and the number of notes per beat are also assigned in the settings. Furthermore, each track in a given piece may use a different chord progression.

The velocity of notes is determined using a similar method to pitch: the time series data is converted into velocities such that the lowest value in the dataset is the quietest value available, and the highest value of the dataset is the loudest value available. The notes in between are assigned proportionally by their data value between the quietest and loudest notes. Each track may have its own customised velocity range, such that any given track may be louder or quieter than the other tracks in a piece. The

choice of dataset used to determine velocity is provided in the settings. We rarely used the same dataset for both pitch and for

velocity. This is because it results in the high pitch notes being louder and the low pitch notes being quieter.

After binning the notes into the appropriate scales, all notes are initially the same duration. If the same pitched note is played successively, then the first note's duration is extended and the repeated notes are removed.

A smoothing function may also be applied to the data before the dataset is converted into musical notes. Smoothing means that it is more likely that the same pitched note will be played successively, so a track with a larger smoothing window will have

200 fewer notes than a track with a smaller window. From the musical perspective, smoothing slows down the piece by replacing fast short notes with longer slower notes. Smoothing can also be used to slow down the backing parts to highlight a faster moving melody. Nearly all the pieces described here used a smoothing window.

After applying this method to multiple tracks, they are saved together in a single MIDI file using the python MIDITime library, (Corey, 2016) Having created the MIDI file, the piece is performed by the TiMidity++ digital piano, (Izumo and

205 Toivonen, 2004), which converts the MIDI format into a digital audio performance in the MP3 format. In principle, it should be possible to use alternative MIDI instruments, but for this limited study we exclusively used the TiMidity++ digital piano. Where possible, the MIDI files were converted into sheet music PDF files using the musescore software, (Musescore BVBA, 2019). However, it is not possible to produce sheet music for all six pieces, as some have too many MIDI tracks to be converted to sheet music by this software.

While the method is relatively straightforward and repeatable, each piece has a diverse range of settings and artistic choices made by the composer: the choice of datasets used to determine pitch and velocity for each track, the pitch and velocity ranges for each track, the piece's tempo and the number of notes per beat, the musical key and chord progression for each track, and the width of the smoothing window. The choice of instrument is also another artistic choice, although in this work, only one instrument was used, the TiMidity+ piano synthesizer. As a whole, these decisions allow the composer to attempt to define

the emotional context of the final piece. For instance, a fast-paced piece in a major progression may sound happy and cheerful to an audience who are used to associating fast-paced songs in major keys with happy and cheerful environments. It should be mentioned that there are no strict rules governing the emotional context of chords, tempo or instrument and the emotional contexts of harmonies, timbres and tempos differ between cultures. Nevertheless, through exploiting the standard behaviours of western musical traditions, the composer can attempt to imbue the piece with emotional musical cues that fit the theme of

the piece or the behaviour of the underlying climate data.

To create a video, we produced an image for each time step in each piece. These figures show the data once they have been converted and binned into musical notes, using units of the original data. A still image from each video is shown in fig. 4. The ffmpeg video editing software, (FFmpeg Developers, 2017), was used to convert the set of images into a video and added the MP3 as the soundtrack.

The finished videos were uploaded onto the lead author's YouTube channel, (de Mora, 2019). The videos were published in an ad hoc manor, as they were available. Each new video was shared via the lead author's personal social media accounts, twitter, Facebook, WhatsApp and the initial playlist was shared on the *data is beautiful* reddit page, (Reddit, 2019). and the

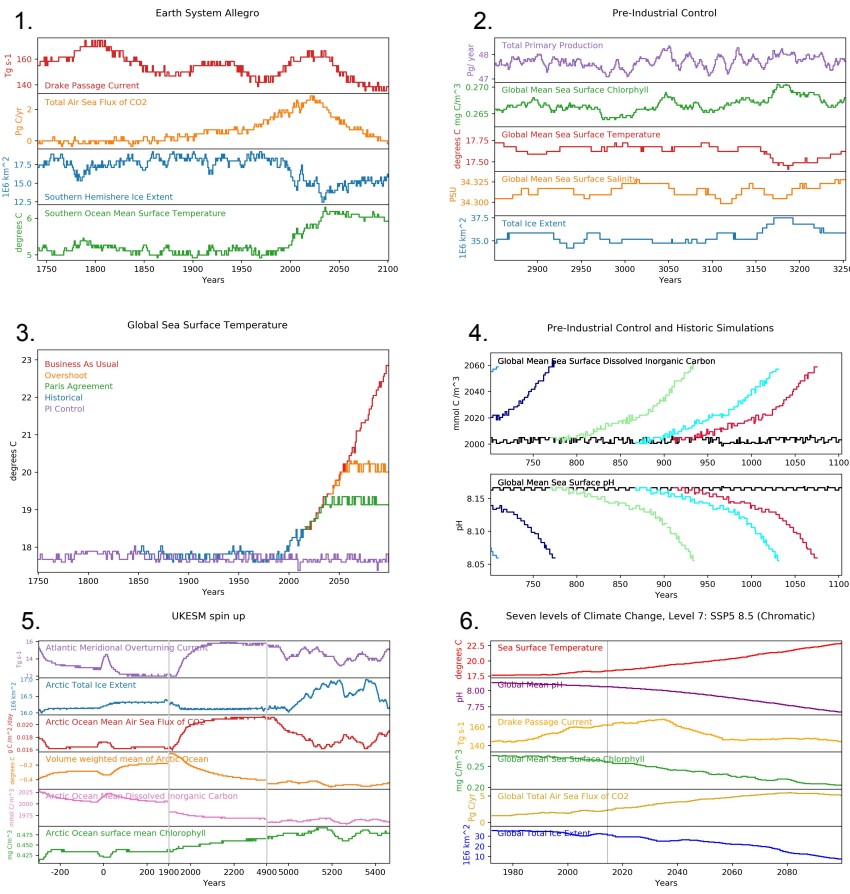

**Figure 4.** The final frame of each of the six videos. These frames of the videos are shown in the order that they were published. The videos 1), 3), 5) and 6) use a consistent x-axis for the duration of the video, but videos 2) and 4) have rolling x-axes that change over the course of the video. This means that panes 2 and 4 show only a small part of time range. Pane 5 includes two vertical lines showing the jumps in the spin up piece. Pane 6 shows a single vertical line for the crossover between the historical and future scenarios.

piano reddit page (Reddit). The videos were also shown at several scientific meetings, notably to an audience of approximately 100 scientists at the National Centre for Earth Observation Annual Conference in September 2019.

 ## 3.1 Works

Six pieces were composed, generated and published using the methods described here. These pieces and their web addresses are:

**Earth System Allegro:** https://www.youtube.com/watch?v=RxBhLNPH8ls

**Pre-industrial Vivace:** https://www.youtube.com/watch?v=Hnkvkx4BMk4

 **Ocean Acidification in E minor:** https://www.youtube.com/watch?v=FPeSAA38MjI

**Sea Surface Temperature Aria:** https://www.youtube.com/watch?v=SYEncjETkZA

**Giant Steps Spin Up:** https://www.youtube.com/watch?v=fSK6ayp4i4w

**Seven Levels of Climate Change:** https://www.youtube.com/watch?v=2YE9uHBE5OI

The main goals of the work were to generate music using climate model data, to use music to illustrate some standard prac-
 tices in Earth System modelling that might not be widely known outside our community, and to quantify the early dissemination of these pieces. Beyond these broader goals, each piece had its own a unique goal: to demonstrate the principles of sonification using UKESM1 data in the *Earth System Allegro*. The *Pre-industrial Vivace* introduces the concept of a pre-industrial control simulation and highlights how an emotional connection can be made between the model output and the sonification of the data. The *Sea Surface Temperature Aria*'s goal was to demonstrate the range of behaviours of the future climate projections. *Ocean*
 *Acidification in E minor* aimed to show the impact of rising atmospheric $CO_2$ on ocean acidification and also to illustrate how historical runs are branched from the pre-industrial control. The *Giant Steps Spin Up* shows the process of spinning up the marine component of the UKESM1, and finally, the *Seven Levels of Climate Change* was aiming to use the musical principles of jazz harmonisation to distinguish the full set of UKESM1's future scenario simulations.

These six pieces are summarised in fig. 4 and tab. 1. Figure 4 shows the final frame of each of the pieces, tab. 1 shows
 the summary information about each of the videos, including the publication date, the duration and lists the experiments and datasets used to generate the piece.

### 3.1.1 Earth System Allegro

The *Earth System Allegro* is a relatively fast-paced piece in C Major, showing some important metrics of the Southern Ocean in the recent past and projected into the future with the SSP1 1.9. The SSP1 1.9 projection is is the future scenario in which
 the anthropogenic impact on the climate is the smallest. The C major scale is composed of only natural notes (no sharp or flat notes), making it one of the first chords that people encounter when learning music. In addition, major chords and scales like C Major typically sound happy. Christian Schubart's 'Ideen zu einer Aesthetik der Tonkunst' (1806) describe C major as "Completely pure. Its character is: innocence, simplicity, naivety, children's talk." As this was the first piece in the series, this seemed an appropriate way to start the Earth System Music project. Through choosing C major and an upbeat tempo, and data

from the best possible climate scenario (SSP1 1.9), we aimed to start the project with a piece with a sense of optimism about the future climate and to introduce the principles of musification of UKESM1 time series data.

The Drake Passage current, shown in red in the top left pane of figure 4, is a measure of the strongest current in the ocean, the Antarctic circumpolar current. This is the current that flows eastwards around Antarctica. The second dataset shown here in orange is the global total air to sea flux of $CO_2$. This field shows the global total atmospheric carbon dioxide that is absorbed into the ocean each year. Even under SSP1 1.9, UKESM1 predicts that this value would rise from around zero during the pre-industrial period to a maximum of approximately 2 Pg of carbon per year around the year 2030, followed by a return to zero at the end of the century. The third field is the sea ice extent of the Southern Hemisphere, shown in blue. This is the total area of the ocean in the Southern Hemisphere which has more that 15% ice coverage per grid cell of our model. The fourth field is the Southern Ocean mean surface temperature, shown in green, which rises slightly from approximately 5 degrees Celsius in the pre-industrial period up to a maximum of 6 degrees. The ranges of each dataset are illustrated in fig. 2.

In this piece, the Drake Passage current is set to the C major scale, but the other three parts module between the C major, G major, A minor and F major chords. These are the first, fifth, sixth and fourth chords in the root of C major. This progression is strikingly popular and may be heard in songs such as: *Let It Be*, by the Beatles, *No Woman No Cry* by Bob Marley and the Whalers, *With or Without You* by U2, *I'm Yours* by Jason Mraz, *Africa* by Toto, among many others. By choosing such a common progression, we were aiming to introduce the concept of musification of data using familiar sounding music and to avoid alienating the audience.

### 3.1.2 Pre-industrial Vivace

The *Pre-industrial Vivace* is a fast-paced piece in C Major, showing various metrics of the behaviour of the Global Ocean in the pre-industrial control run. The pre-industrial control run is a long term simulation of the Earth's climate without the impact of the industrial revolution or any of the subsequent human impact on climate. At the time that the piece was created, there were approximately 1400 simulated years. We use the control run as starting points for historical simulations, but also to compare the difference between human-influenced and simulations of the ocean without any anthropogenic impact.

The final frame of the *Pre-industrial Vivace* video is shown in the top right pane of figure 4. The top pane of this video shows the global marine primary production in purple. The primary production is a measure of how much marine phytoplankton is growing. Similarly, the second pane shows the global marine surface chlorophyll concentration in green; this line rises and falls alongside the primary production in most cases. The third and fourth panes show the global mean sea surface temperature and salinity in red and orange. The fifth pane shows the global total ice extent. These five fields are an overview of the behaviour of the pristine natural ocean of our Earth System model. There is no significant drift and there is no long term trend in any of these fields. However, there is significant natural variability operating at decadal and millennial scales.

As with the *Earth System Allegro*, *Pre-industrial Vivace* uses the familiar C major scale but adds a slight variation to the chord progression. The first half of the progression is C major, G major, A minor and F major, but it follows with a common variant of this progression: C major, D minor, E minor and F major. Through using the lively vivace tempo and a familiar chord progression in a major key, this piece aims to use musification to link the pre-industrial control simulation with a sense of

happiness and ease. The lively, fast, jovial tone of the piece should match the pre-industrial environment which is free running
and uninhibited by anthropogenic pollution.

### 3.1.3  Sea Surface Temperature Aria

The *Sea Surface Temperature Aria* demonstrates the change in the sea surface temperature in the pre-industrial control run,
the historical scenario and under three future climate projection scenarios, as shown in pane 3 of fig. 4. The three scenarios
are the "business as usual'" scenario, SSP5 8.5, where human carbon emissions continue without mitigation shown in red. The
second scenario is an "overshoot" scenario, SSP5 3.4-overshoot, where emissions continue to grow, but then drop rapidly in
the middle of the $21^{st}$ century, shown in orange. The third scenario is SSP1 1.9, labelled as the "Paris Agreement" scenario,
where carbon emissions drop rapidly from the present day, shown in green. The goal of this piece is to demonstrate the range
of differences between some of the SSP scenarios on Sea Surface Temperature.

The pre-industrial control run and much of the historical scenario data are relatively constant. However, they start to diverge
in the 1950s. In the future scenarios, the three projects all behave similarly until the 2030s, then the SSP1 1.9 scenario branches
off and maintains a relatively constant global mean sea surface temperature. The SSP5 3.4 scenario's SST continues to grow
until the year 2050, while the SSP5 8.5 scenario's SST grows until the end of the simulation.

Musically, this piece is consistently in the scale of A minor harmonic with no chord progression. The minor harmonic scale
is a somewhat artificial scale in that it augments $7^{th}$ note of the natural minor scale. The augmented $7^{th}$ means that there
is a minor third between the $6^{th}$ and $7^{th}$ note, making it sound uneasy and sad (at least to the author's ears). An aria is a
self-contained piece for one voice, normally within a larger work. In this case, the name aria is used to highlight that only one
dataset, the sea surface temperature, participates in the piece. This piece starts relatively low and slow, then grows higher and
louder as the future scenarios are added to the piece. The unchanging minor harmonic chord, slow tempo and pitch range were
chosen to elicit a sense of dread and discord as the piece progresses to the catastrophic SSP5 8.5 scenario at the end of the $21^{st}$
century.

### 3.1.4  Ocean acidification in E minor

*Ocean acidification in E minor* demonstrates the standard modelling practice of branching historical simulations from the pre-
industrial control run, as well as the impact of rising anthropogenic carbon on the ocean carbon cycle. The final frame of this
video is shown in pane 4 of fig. 4. The top pane shows the global mean dissolved inorganic carbon (DIC) concentration in the
surface of the ocean and the lower pane shows the global mean sea surface pH. In both panes, the pre-industrial control run
data is shown as a black line and the coloured lines represent the fifteen historical simulations.

This piece uses a repeating *12-bar blues* structure in E minor and a relatively fast tempo. This chord progression is an
exceptionally common progression, especially in blues, jazz and early rock and roll. It is composed of four bars of the E minor,
two bars of A minor, 2 bars of E minor, then one bar of B minor, A minor, E minor and B minor. The twelve bar blues can be
be heard in songs such as: *Johnny B. Goode* by Chuck Berry, *Hound Dog* by Elvis Presley, *I got you (I feel Good)* by James
Brown, *Sweet Home Chicago* by Robert Johnson or *Rock n Roll* by Led Zeppelin. In the context of Earth System Music, the

12-bar pattern with its opening set of four bars, then two sets of two bar and ending for four sets of one bar between key changes drives the song forward before starting again slowly. This behaviour is thematically similar to the behaviour of the ocean acidification in UKESM1 historical simulation, where the bulk of the acidification occurs at the end of each historical period.

This video highlights that the marine carbon system is heavily impacted over the historical period. In the pre-industrial control runs, both the pH and the DIC are very stable. However, in all historical simulations with rising atmospheric $CO_2$, the DIC concentration rises and the pH falls. The process of ocean acidification is relatively simple and well understood (Caldeira and Wickett, 2003; Orr et al., 2005). The atmospheric $CO_2$ is absorbed from the air into the surface ocean, which releases hydrogen ions into the ocean, making the ocean more acidic. The concentration of DIC in the sea surface is closely linked with the concentration of atmospheric $CO_2$, and it rises over the historic period. This behaviour was observed in every single UKESM1 historical simulation.

This video also illustrates an important part of the methodology used to produce models of the climate that may not be widely known outside our community. When we produce models of the Earth System, we use a range of points of the pre-industrial control as the initial conditions for the historical simulations. All the historical simulations have slightly different starting points, and evolve from these different initial conditions, which gives us more confidence that the results of our projections are due to changes since the pre-industrial period instead of simply a consequence of the initial conditions. In this figure, the historical simulations are shown where they branch from the pre-industrial control run instead of using the "real" time as the x-axis.

### 3.1.5   Giant Steps Spin Up

This piece combines the spin up of the United Kingdom Earth System Model with the chord progression of John Coltrane's *Giant Steps*, (Coltrane, 1960). The spin up is the process of running the model from a set of initial condition to an equilibrium steady state. When a model reaches a steady state, this means that there is no significant trend or drift in the mean behaviour of several key metrics. For instance, as part of the C4MIP protocol, Jones et al. (2016) suggest a drift criterion of less than 10 Pg of carbon per century in the absolute value of the flux of $CO_2$ from the atmosphere to the ocean. In practical terms, the ocean model is considered to be spun up when the long-term average of the air sea flux of carbon is consistently between -0.1 and 0.1 Pg of carbon per year.

The spin up is a crucial part of model development. Without spinning up, the historical ocean model would still be equilibrating with the atmosphere. It would be much more difficult to separate the trends in the historical and future scenarios from the underlying trend of a model still trying to equilibrate. Note that while a steady state model does not have any significant long term trends or drifts; it can still have short term variability. This short term variability can be seen in the pre-industrial simulation in the *Pre-industrial Vivace* piece. It can take a model thousands of years of simulation for the ocean to reach a steady state. In our case, the spin up ran for approximately 5000 simulated years before the spun up drift criterion were met (Yool et al., 2020).

The UKESM1 spin up was composed of several phases in succession. The first stage was a full coupled run using an early version of UKESM1. Then, an ocean-only run was started using a 30 year repeating atmospheric forcing dataset. The beginning of this part of the run is considered to be the beginning of the spin up and the time axis is set to zero at the start of this run. This is because the early version of UKESM1 did not include a carbon system in the ocean. After about 1900 years of simulating the ocean with the repeating atmospheric forcing dataset, we had found that some changes were needed to the physical model. At

this point, we initialised a new simulation from the final year of the previous stage and changed the atmospheric forcing. This second ocean-only simulation ran until the year 4900. At the point, we finished the spin up with a few hundred years of fully coupled UKESM1, with ocean, land, sea ice and atmosphere models. Due to the slow and repetitive native of the ocean-only spin up, several centuries of data were omitted. These are marked as grey vertical lines in the video and in the bottom left pane of fig. 4.

The piece is composed of several important metrics of the spin up in the ocean, such as the Atlantic meridional overturning current (purple), Arctic ocean total ice extent (blue), the global air sea flux of $CO_2$ (red), the volume weighted mean temperature of the Arctic ocean (orange), the surface mean DIC in the Arctic Ocean (pink) and the surface mean chlorophyll concentration in the Arctic ocean (green).

The music is based on the chord progression from the jazz standard, John Coltrane's *Giant Steps*, although the musical

progression was slowed to one chord change per four beats instead of a change every beat. This change occurred as an accident, but we found that the full speed version sounded very chaotic, so the slowed version was published instead. This piece was chosen because it has a certain notoriety due to the difficulty for musicians to improvise over the rapid chord changes. In additional, *Giant Steps* was the first new composition to feature Coltrane changes. Coltrane changes are a complex cyclical harmonic progression, which forms a musical framework for jazz improvisation. We hoped that the complexity of the Earth

system model is reflected in the complexity of the harmonic structure of the piece. The cyclical relationship of the Coltrane changes also reflects the 30 year repeating atmospheric forcing dataset used to spin up the ocean model.

### 3.1.6 Seven Levels of Climate Change

This piece is based on a YouTube video by Adam Neely, called The 7 Levels of Jazz Harmony, (Neely, 2019). In that video, Neely demonstrates seven increasingly complex levels of jazz harmony by re-harmonising a line of the chorus of Lizzo's song

*Juice*. We have repeated Neely's re-harmonisation of *Juice* here, such that each successive level's note choice is informed by Earth System simulations with increasing levels of emissions and stronger anthropogenic climate change.

At the time of writing, UKESM1 had produced simulations of seven future scenarios. The seven scenarios of climate change and their associated Jazz harmony and are:

- Level 0 : Pre industrial control - Original Harmony

- Level 1 : SSP1 1.9 - 4 note chords

- Level 2 : SSP1 2.6 - Tritone substitution

- Level 3 : SSP4 3.4 - Tertiary harmony extension

- Level 4 : SSP5 3.4 (overshoot) - Pedal Point

- Level 5 : SSP2 4.5 - Non-functional harmony

- Level 6 : SSP3 7.0 - Liberated dissonance

- Level 7 : SSP5 8.5 - Fully chromatic

Note that we were not able to reproduce Adam's seventh level: intonalism or xenharmony. In this level, the intonation of the notes are changed depending on the underlying melody. Unfortunately, the MIDITime python interface to MIDI has not yet reached such a level of sophistication. Instead, we simply allow all possible values of the 12 note chromatic scale.

The datasets used in this piece are a set of global scale metrics that show the bulk properties of the model under the future climate change scenarios. They include the global mean SST (red), the global mean surface pH (purple), the Drake Passage current (yellow), the global mean surface chlorophyll concentration (green), the global total air to sea flux to $CO_2$ (gold) and the global total ice extent (blue). As the piece progresses through the seven levels, the anthropogenic climate change of the model becomes more extreme, matching the increasingly esoteric harmonies of the music.

**3.2 Quantification of reach**

YouTube's built-in toolkit for channel monitoring, YouTube studio, was used quantify the reach, engagement and audience for the individual videos, as well as the entire channel (Google, 2019). Using these tools, we have investigated the first 90 days that the videos were published, starting from the $21^{st}$ of August 2019 to the $18^{th}$ of November. This is because YouTube studio can only produce reliable data on unique viewers for a period up to 90 days. Unique viewers are an estimate of the total number of

410 people to watch a video. This metric accounts for when a single person watches a video on multiple devices or several times on the same device. It also accounts for when multiple people share the same device using separate YouTube accounts.

In contrast to the unique viewers metric, the view count metric includes views from viewers who re-watch the same video multiple times. Of course, neither unique viewers or view count can account for the audience size if multiple people watch a video at the same time on the same screen or if the video is shown to larger audiences.

Using YouTube studio, it is also possible to retrieve some basic demographic information about the viewers: where they are in the world, their gender, age and how they came across the video (traffic source). However, this channel has too few views to get information about gender and age, so we are limited to geography and traffic source.

These videos were published as they were ready and shared in an ad hoc way via the lead author's personal Facebook, twitter, WhatsApp and reddit social media networks. The sharing posts included a link to the new video, and a brief comment on the

420 contents of the video. Each video was posted alongside a brief description and used the three main tags: #Music, #Science and #ClimateChange, as well as several other minor hashtags. Each piece was also added to a YouTube playlist, which was shared via social media in the same way as the individual pieces. The playlist lists the pieces in the order that they were published. As this was a pilot study, there were no particular timed-releases, no press release and no direct assistance from the PML, NERC

or UKESM communications teams. The videos were then disseminated through these networks and allowed to reach wider audiences. In addition, no paid advertisements were purchased. There is very little publicly available data about the individuals who have shared these videos within private networks such as Facebook or WhatsApp. Furthermore, it is not possible to know how many people viewed their posts. For this reason, this work focuses on the data made available to content creators in YouTube studio. Please see sect. 6 for a discussion on possible extensions to this pilot study where these avenues could be explored.

## 4 Results

Table 2 shows a summary of the reach of these videos including the number of unique viewers, the total number of views, the average view duration, the average audience retention and the cumulative watch time for each of the six videos and a total. The number of unique viewers is approximately half of the total number of viewers which suggests that many viewers watched the videos several times. The average view duration indicates how long the viewers watched the video and the average audience retention is the percentage of the video that the average audience member viewed.

The cumulative number of unique viewers for each of the six videos over the first 90 days of the project is shown in fig. 5, and fig. 6 shows the cumulative non-unique view count for each video. The x axis of these figures has been zeroed such that the day that the first video was released (21/08/2019) is day zero. These figures both show that the keenest interest in the videos was at the beginning of the project, when three videos were published in quick succession. The later three videos did not receive the same number of views. After the initial period of high interest, nearly all videos have a similar viewing rate, approximately 5 views per month. Figure 7 shows the total number of views on the day that each video was published. It is worth noting that the first two videos were both published on the same day. The *Earth System Allegro* had the highest opening day view count, and the *Sea Surface Temperature Aria* video had the lowest opening day view count.

YouTube studio also provides the geographic distribution of part of the viewership, based on the IP address of the viewer. The origin of more than half of the viewership is unknown due to the restriction that content creators may only see demographics data for a subset of viewers. Outside of the viewers of unknown origin (56.6%), the United Kingdom dominates the list with 32.5%, followed by the United States (6.5%), France (2.2%) and Germany (2.2%).

Figure 8 shows the source of the total views for this channel divided into categories. The traffic sources are external: traffic from websites and apps that have the embedded or linked the video; playlist: traffic from any playlist that included one of the videos; playlist page: traffic from the Earth System music playlist page; channel page: traffic from the YouTube channel page; direct: external traffic from direct URL entry; YouTube search: traffic from the YouTube search engine; other YouTube: traffic from other places in YouTube, for instance from subscribers and notifications; browse features: traffic from the home screen, subscription feed, watch later and other browsing features; and finally, suggested videos: traffic from the suggestions that appear next to or after other YouTube videos.

This figures clearly shows that the largest source of the traffic is from External sources, which is likely to be from people watching the videos on social media such as Facebook, Twitter or reddit. Using YouTube studio, it is not possible to further

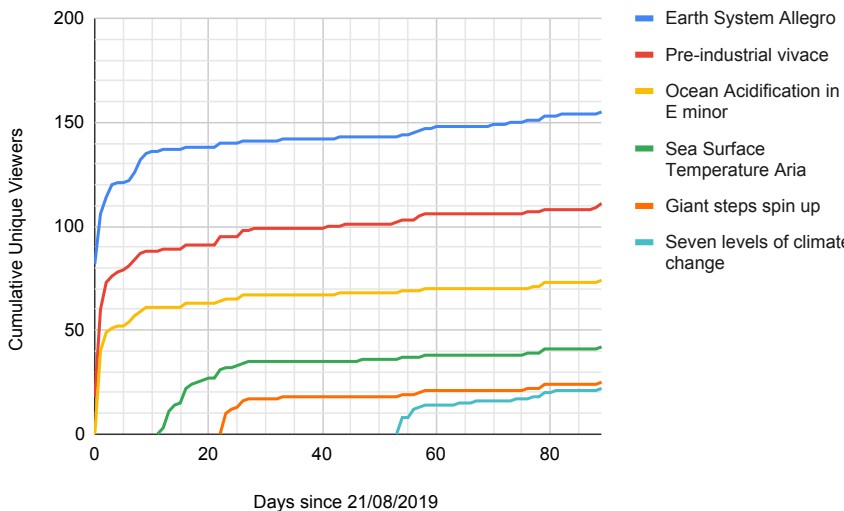

**Figure 5.** The cumulative number of unique viewers of each video over the 90 day period starting from the publication of the first videos.

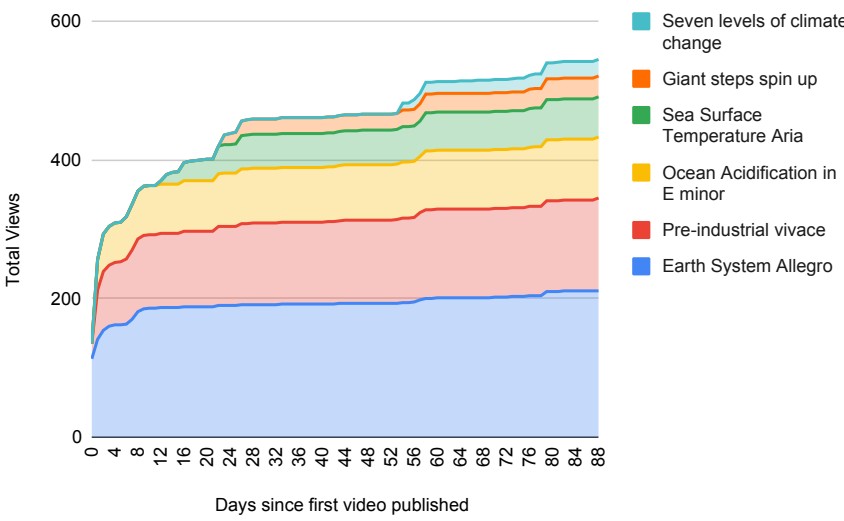

**Figure 6.** The total cumulative non-unique view count of each video over the 90 day period starting with the publication of the first video.

differentiate which external source led to the view. The second largest single source of traffic is the playlist, which means that many viewers watched several videos in a row from the playlist. There is relatively little traffic from within YouTube in the suggested videos and browse features traffic sources. This suggests that most of the views come from people within the lead author's social media network, and that YouTube has not included these videos in suggestions to a wider audience.

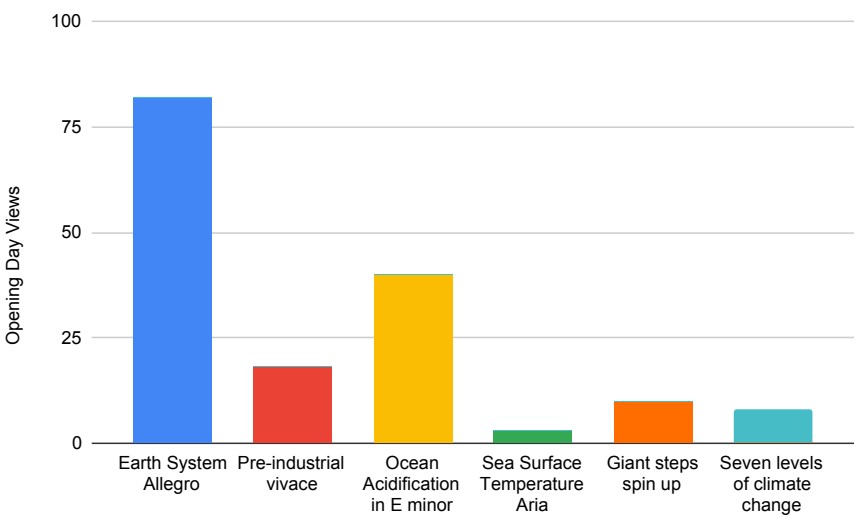

**Figure 7.** The total view count of each video on the day that it was published.

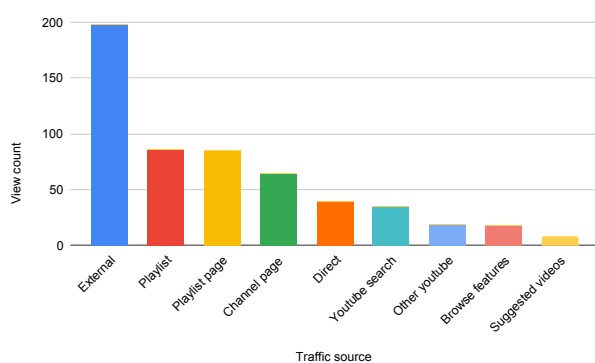

**Figure 8.** The traffic source the views of these videos. This figure shows how viewers came to watch these videos.

Aside from the metrics shown in tab. 2, YouTube studio provides a few other metrics of channel behaviour. For instance, during this time period, the number of channel subscribers rose from 1 to 10, and 8% of the view time came from subscribers. The 6 videos received a total of two comments, ten likes and zero dislikes on YouTube.

The following statement were posted via social media. These were posted directly on the YouTube video page, or on social media posts linking to the video via facebook, twitter and reddit. Note that I have removed emojis and gifs, but otherwise reproduced comments as they originally appeared.

1. This one was very dramatic.

2. It gets quite dramatic after 1950-60

3. This was submitted as a scientific research paper here, but don't understand what the point is.

4. Wow! That's awesome!

5. Great idea

6. That's Crazy!!

7. Awesome! Personally I was hoping to hear something of the same ilk as system of a down or similar. Does make modelling sound far more upbeat though

8. Brilliant idea! Just like Herman Hess's book 'The Glass bead Game', where data is unified from many sources and brought together by many senses. Listen to how the ocean sings!

9. AMAZING!!!!

10. A total new meaning for the "listen to the ocean" motto

11. That's awesome! Can you post this to someone with a bigass twitter handle. This deserves more attention. Maybe send it to Adam Rutherford at Inside Science?

12. It's quite different from the previous one! Super Cool stuff!!

13. This is amazing. If you can find a setup that would cause noticeable change in the music between pre-industrial and future you're viral

14. I love this!! Well done!

15. This is amazing ! Was initially quite surprised that these weren't more tonally chaotic - then I read I the blurb for the top video. I presume that you could do the same with any choice of key and scale? Also, I hope you realise that, with your choice of C-major, you have made something worthy on inclusion in the next Axis of Awesome medley

16. Certainly very interesting, but it lacks a lot of human touch. These generations don't seem to take into account many compositional techniques that are almost vital to make a comprehensive piece of music, such as motifs, dynamics, musical form, things like that. I found that there wasn't much for the ear to catch onto musically, no clear melodies, rhythms, harmonic progressions etc. Because of this it can also be quite hard for a human to learn, adding on to the challenge of coordinating the strange rhythms between the hands. So it's cool, but very unhuman music and thus most likely difficult for humans to play.

17. I think if humans would take their time to learn this, the musical phrasing they would bring to the table could do a lot to make the pieces easier and more enjoyable to listen to. I think computer made music is very cool in that it really

highlights the amount of complexity that goes into creating music as we would know it, namely through the human mind. So the kind of research you're conducting is very valuable indeed!

18. Wow this actually sounds very interesting! Sadly I'm not a very accomplished pianist either, but I'll leave this comment and upvote for visibility.

## 5  Discussion

The main goals of the work were to generate music using climate model data, to use music to illustrate some standard practices in Earth System modelling that might not be widely known outside our community, and to quantify the early dissemination of these pieces. Six pieces were generated, each piece used UKESM1 data, and the included a wide range of ocean model behaviour and methods. These pieces reached an audience of 251 people and were viewed 553 times in the first 90 days that they were published. Approximately half of views occurred in the first week that the videos were published. The views per day decreased after this initial two week period. Each new video received fewer total views than the previous videos during this initial period.

The *Earth System Allegro* was the first video to be posted, and it received the most views, the largest number of total views, the highest opening day views, the most watch time and the most likes. The *Seven Levels of Climate Change* piece was the last video to be posted and received the least opening day views but also the least total views. In addition, each new video received fewer total views than the previous video over the entire 90 day time scale as well as during the initial period. This may be partially explained by each new video that was published typically resulted in the other videos being viewed as well, so new videos are unlikely to overtake older videos over their lifetime.

The piece that had the highest audience retention was the *Sea Surface Temperature Aria*. This is not surprising either, as the end is arguably the most interesting part of this piece. This pieces starts slowly with a century of pre-industrial control by itself, then the historical dataset is added for another 165 years. In the final section of this video, the future scenarios diverge from the historical and pre-industrial control run. The global mean SST in scenarios rises rapidly in unison over the early part of the $21^{st}$ century, then diverge in the second half of the century. The *Sea Surface Temperature Aria* is also the most visually simple animation of the six pieces. Only one pane is visible in the video and much of the piece only includes one or two voices at a time. It may be possible that this simplicity helps to hold the audience's attention. This piece is the work that is most similar to the other climate change music pieces like (Borromeo et al., 2016) and (Crawford, 2013), described in sect. 1. In addition, this piece had the lowest opening day view count, but did not have the lowest total view count at the end of the 90 day time range.

The total number of views of the Earth System music playlist is very small when compared to the most popular YouTube channels, which may have millions of subscribers and views. However, when contrasted against other videos on Earth System modelling, the Earth System Music playlist has a comparable total view count. As a comparison, the *UKESM short video introduction*, https://www.youtube.com/watch?v=hclsFbnmUdI, video was published in the CRESCENDO H2020 project YouTube channel on the $1^{st}$ of November 2017 and has received 183 views and zero likes or dislikes in the first two years since it was published. This is the only other video about UKESM1 on YouTube, and the Earth System Music playlist has re-

ceived three times the total views in the first 90 days of the project. By itself, the Earth System Allegro video received more views in the first fortnight than the *UKESM short video introduction* received in two years. However, the YouTube video about Earth System Models with the largest viewership, *E3SM: DOE's New, State-of-the-Science Earth System Model*, https://www.youtube.com/watch?v=8Df96rx3i9g, has received 4479 views after 18 months online. Similarly, the NERC science YouTube channel has more than 700 subscribers at the time of submission, the most popular video *Anatomy of an earthquake - Professor Iain Stewart* has 82K views, (NERC and Stewart, 2014). Both these videos have received significantly more views than the Earth System playlist.

The geographic distribution of the viewership based on the IP address was also included. This work originates in the United Kingdom and uses data from the United Kingdom Earth System Model, so it is not surprising that the United Kingdom is the largest source of viewers. It is likely that these works have been viewed outside those four countries, however no data is provided for 56.5% of the views, due to the restrictions imposed by YouTube.

Outside of the viewership statistics, a small number of people were sufficiently interested to subscribe to the channel or to like a video. Subscribing to a YouTube channel means that the future videos from that channel are more likely to appear on your main YouTube page.

This project was focused on demonstrating that we could use Earth System model data to generate music and share it. While we hoped to improve the wider public's understanding of the methods used in climate change modelling, the tools available to us within YouTube studio do not allow any way of assessing this.

The comments from social media were listed above in sect. 4 and were almost all positive and supportive. However, these comments are biased towards the author's friends, family and professional colleagues. These comments include several positive comments and praise, comments about the pieces themselves ("it gets dramatic at 1950-1960"), comparisons to other musicians or works of art ( e.g. System of a Down, Herman Hess's book *The Glass Bead Game*, the Axis of Awesome medley ). One person "didn't understand what the point is", and the final three comments contain some interesting insight into the musical side of the work.

## 6   Limitations and Future Work

As mentioned earlier, the main goals of this pilot study were to generate music using climate model data, to use music to illustrate some standard practices in Earth System modelling that might not be widely known outside our community, and to quantify the early dissemination of these pieces. We have successfully demonstrated that it is possible to generate music using our climate model's data. It is less conclusive whether we can teach the wider community about the methods of climate modelling. We have also found that the YouTube studio toolkit is not currently sufficiently capable to fully quantify the reach of these videos. We also make several suggestions for methods to reach a wider audience through improving the quality of the videos.

While we hoped to disseminate information about Earth System modelling to a wider audience, it is not possible to determine whether the audience learned anything about Earth System modelling using the metrics provided by YouTube studio or the

comments posted on social media. Furthermore, it is not possible to determine whether the audience was composed of laymen or experts. As this was a pilot study, we did not go into greater detail to understand the audience reactions. Future extensions of this project should include a survey of the audience, investigating their backgrounds, demographics, what they learned about Earth System models and their overall impressions of the pieces. This could take the form of an online survey associated with each video, or a discussion with the audience at a live performance event.

Our videos only include the music and a visualisation of the data, they do not include any description about how the music was generated or the Earth system modelling methods used to create the underlying data. The explanations of the science and musification methodologies are held in a text description below the video. Furthermore, viewers must expand this box by clicking the "show more" button. Using YouTube studio, it is not currently possible to determine whether the viewers have expanded, read or understood the description section. When we have shown these videos to live audiences at scientific meetings and conferences, it has always been associated with a brief explanation of the methods. In the future, this explanatory preface to the work could be included in the video itself or as a separate video, as well as below the video in the description section.

If additional pieces were made, there are several potential ways that these could improve over the current set of videos. In future versions of this work, it should be possible to use ESMValTool (Righi et al., 2019) to produce the time series data instead of BGC-val. This would make the production of the time series more easily repeatable, but also would also make it easier for pieces to be composed using data available in CMIP5 and CMIP6 coupled model intercomparison projects. This broadens the scope of data by allowing other models, other model domains including the atmosphere and the land surface, and even observational datasets. For instance, we could make a multi-model intercomparison piece, or a piece based on the atmospheric, terrestrial and ocean components of the same model. In addition, using ESMValTool would also make it more straightforward to distribute the source code that was used to make these pieces.

In their reflections on auditory graphics, Flowers (2005) lists several "Things that work" and "Approaches that do not work". From the list of things that work, we included four of the five methods that worked: pitch coding of numeric data, the exploitation of temporal resolution of human audition, manipulating loudness changes, and using time as time. We were not able to include the selection of distinct timbres to minimise stream confusion. From the list of approaches that do not work, we successfully avoided several of the pitfalls, notably pitch mapping to continuous variables, using loudness changes to represent an important continuous variable. However, we did include one of the approaches that Flowers did not recommend: we simultaneously plot several variables with similar pitches and timbres. However, it is worth noting that maximising the clarity of the sonification is the goal of Flowers (2005), but our focus was to produce and disseminate some relatively listenable pieces of music using UKESM1 data.

The two Flowers (2005) suggestions that we failed to address were both related to using the same timbre digital piano synthesizer for all data. Due to the technical limitations of using TiMidity++, we were not able to vary to the instruments used, and thus there was very little variability in terms of the timbres. These pieces were all performed by the same instrument, a solo piano, which limits the musical diversity of the set of pieces. In addition, each dataset within in a given piece was performed by the same instrument, making it difficult to distinguish the different datasets being performed simultaneously. Further extensions of this work could use a fully featured digital audio workstation to access a range of digital instruments

beyond the digital piano, such as a string quartet, a horn and woodwind section, a full digital orchestra, electric guitar and bass, percussive instruments, or electronic synthesised instruments. This would comply with the suggestions listed in Flowers (2005), allowing the individual datasets to stand out musically from each other in an individual piece, but would also lead to a

much more diverse set of musical pieces.

From a musical perspective, there are many ways to improve the performances of the pieces for future versions of this work. As raised in the comments from social media, a human pianist would be able to add a warmth to the performance that is beyond the abilities of MIDI interpreters. A recording of a human performance could also add the hidden artefacts of live recording, such as room noise, stereo effects, and natural reverb. On the other hand, due to the nature of the process used to generate

these pieces, it is possible that it may not be possible for a single human to perform several of the pieces due to the speed, complexity, number of simultaneous notes or the range of these pieces. Alternatively, it may be possible to "humanise" the MIDI by making subtle changes to the timing and velocities of the MIDI notes. This is a recording technique that can take a synthesised perfectly timed beat and make it sound like it is played by a human. It does this by moving the individual notes slightly before or after the beat, and adding subtle variations in the velocity (John Walden, 2017). Also, TiMidity++ uses the

same piano sample for each pitch. This means that when two tracks of a piece play the same pitch at the same time, the exact same sample is played twice simultaneously. These two identical sample sound waves are added constructively and the note jumps out much louder than it would be if a human played the part. A fully featured digital piano or a human performance would remove these loud jumps, but also be able to add more nuance and warmth to the performance. Finally, the published pieces had no mastering or post-production. Even a basic mastering session by a professional sound engineer would likely

improve the overall quality of the sound of these pieces.

In terms of the selection of chords progression, tempo, and rhythms, it may be possible to target specific audiences using music based on popular artists or genres. For instance, the reach of a piece might be increased by responding to viral videos or by basing a work on a popular trending song.

In these works, we have focused on reproducing western musical, both traditional and modern, in order to connect each piece

with the associated emotional musical cues. Alternatively, there is a significant diversity of traditional and modern styles of music from every country in the world; a much wider range of rhythms, timbres, styles and emotional cues could be exploited in future extensions of this work.

With regards to the visual aspect of these videos, it should be straightforward to improve the quality of the graphics used. The current videos only show a simple scalar field as it develops over time. They could be improved by adding animated global

maps of the model, interviews or live performances to the video. It may also be a positive addition to preface the videos with a brief explanation of the project and the methods deployed. On the technical side, there may also be some visual glitches and artefacts which arise due to YouTube's compression or streaming algorithms. A different streaming service or alternative video making software might help remove these glitches.

We found that the bulk of the views originated from external links and direct links. This means that the sharing the videos

over social media dominated over YouTube's in-built video suggestions. While it was beyond the scope of this trial project, in future projects it might be possible to change the balance of external to internal traffic and increase the reach of this work

through paid advertising on YouTube and other social media platforms. This would place the videos higher in the suggested video rankings and on the discovery queues.

YouTube videos are typically shown in the suggestions queue with a thumbnail image and the video title. The thumbnail is the graphic placeholder that shows the video while it is not playing, on YouTube as a suggested video, or in the Facebook or Twitter feeds. The thumbnail is how viewers first encounter the video and it is a crucial part of attracting an audience. There are lots of guides helping create better thumbnails (Kjellberg and PewDiePie, 2017; Video Influencers, 2016; Myers, 2019). Future works should attempt to optimise the video thumbnail to attract a wider audience.

In terms of the reach, the authors expect there to be a modest increase in the view count upon the publication of this work. However, judging from the history of the video's audience size, and other videos in this field, we do not expect a significant change in the total number of views. If the goal of future projects were to increase the audience size, then it might be possible to reach a wider audience using a press release, a live performance, a public showing of the videos, or through a collaboration with other musicians or YouTube content creators. It may also be possible to host a live concert, make a live recording, or broadcast a YouTube live stream. It is not fully understood how a video can go viral (West, 2011; Jiang et al., 2014). However, view counts can rise exponentially when a single person or organisation with a large audience shares a video. Improvements to the music, the video, the description and the thumbnail make it more likely for an influencer to like, share, or re-tweet a piece, which could result in an significant increase in the audience size and view count.

The videos in this work were posted to YouTube in an ad hoc fashion, as soon as they were finished. To maximise the number of views, online guides recommend consistent, scheduled in advance, weekly videos, and it's been advised to publish them late in the week in the afternoons (Katie Nohr, 2017; Think Media, 2017).

## 7 Conclusions

In this work, we took data from the first United Kingdom Earth System Model and converted it into six musical pieces and videos. These videos were posted on a YouTube channel and shared via the lead author's personal social media network. In the first 90 days of the videos being published, they reached an audience of 251 unique viewers and were viewed a total of 553 times. The viewers originated in at least four countries and largely got to the channel from a direct shared link of the video or the playlist. Approximately half of views occurred in the first week that the videos were published and the views per day decreased after this point. Each new video that was published typically resulted in the other videos being viewed as well.

Due to the way that these videos were disseminated through the lead authors personal and professional social networks, most of these 251 unique viewers will likely be familiar with the field of climate change research. However, it is less likely that the audience will be familiar with the core principles of climate modelling or ocean modelling: pre-industrial control runs, the spin up process, the multiple future scenarios, the Drake Passage current, the air sea flux of $CO_2$ or the Atlantic meridional overturning circulation. These standard tools in the arsenal of climate modelling are not yet widely appreciated outside our community. These six musical pieces open the door on a new, exciting and fun approach to how we engage with the wider public.

We have also discussed some ways to improve future iterations of this pilot study. To extend the reach, future works could be performed to a live audience, we could collaborate with musicians, and the viewership would likely be increased with improved video graphics, thumbnails, live performances, video diversity, and more frequent upload rates. The scientific content of the videos could be expanded by accessing new datasets, other parts of the UKESM1 Earth System, other CMIP models, or observational datasets. The quality of the music could be improved by including additional instruments and musical genres, and by making live recordings instead of MIDI performance. The knowledge transfer aspect of the project could be improved upon by appending explanations of the science to the video, and by surveying the audience to identify the impact of these works.

Finally, the authors would like to encourage other scientists to think about how their work may be sonified. You may have beautiful and unique music hidden within your data; the methods described in this work would allow it to be made manifest.

*Data availability.* The sheet music and the MIDI files are available alongside this publication.

*Video supplement.* These videos are published online in the YouTube channel: https://www.youtube.com/c/LeedeMora.

*Author contributions.* LdM used BGC-val to produce the model time series data, sonified the BGC-val data, published the videos, performed the analysis of the reach and prepared the text. AAS, RSS and JW provided feedback and early discussions on music in ESM, AY, JP, TK helped develop the core time series data sets in UKESM1, RJP shared the finished videos and provided audience feedback, JCB and CGJ lead the PML modelling group and UKESM1 projects, respectively and both provided crucial feedback and support.

*Competing interests.* Like most YouTube content creators, L de Mora has a financial relationship with YouTube. However, at the time of writing, the channel in which these videos were posted did not meet YouTube's monetisation requirements (1000 subscribers and 4000 hours watched).

*Acknowledgements.* LdM, AY, JP, RSS, TK, RP, JCB and CGJ were supported by the National Environmental Research Council (NERC) National Capability Science Multi-Centre (NCSMC) funding for the U.K. Earth System Modelling project. AAS and JW were supported by the Met Office Hadley Centre Climate Programme funded by BEIS and Defra. CGJ, TK, and RSS through Grant NE/N017978/1; RP through Grant NE/N018079/1; and AY, JP, LdM and JCB through Grant NE/N018036/1. CGJ, TK, AY, JP additionally acknowledge the EU Horizon 2020 CRESCENDO project, Grant 641816.

We acknowledge use of the Monsoon2 system, a collaborative facility supplied under the joint Weather and Climate Research Programme, a strategic partnership between the Met Office and the Natural Environment Research Council.

The simulation data used in this study are archived at the Met Office and are available for research purposes through the JASMIN platform (www.jasmin.ac.uk) maintained by the Centre for Environmental Data Analysis (CEDA).

The authors would also like to thank anyone that took the time to watch a video, leave a comment, use the like button, subscribe to the channel or share these videos.

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

**Table 1.** Table showing the number of unique viewers, total views, the average view duration and the time spent watching each video. This table also includes YouTube's unique video identifier for each video.

| Video title | Publication date | Duration, Minutes:seconds | Experiments | Datasets |
|---|---|---|---|---|
| *Earth System Allegro* RxBhLNPH8ls | 21-08-2019 | 1:02 | Historical, SSP1 2.5 | Drake Passage current, Total Air sea flux of $CO_2$, Southern Hemisphere ice extent, Southern Ocean SST |
| *Pre-industrial Vivace* Hnkvkx4BMk4 | 21-08-2019 | 2:27 | PI Control | Total Primary Production, Global mean sea surface ch SST, SSS, Total ice extent |
| *Ocean Acidification in E minor* FPeSAA38MjI | 22-08-2019 | 1:56 | PI control, historical | Global mean surface DIC, Global mean surface mean pH |
| *Sea Surface Temperature Aria* SYEncjETkZA | 02-09-2019 | 1:17 | PI control, historical, SSP1 1.9, SSP5 3.4 OS, SSP5 8.5 | Global mean SST |
| *Giant Steps Spin Up* fSK6ayp4i4w | 13-09-2019 | 2:52 | Spin up | Atlantic meridional overturning current, Arctic Ice exte Arctic mean air sea flux of $CO_2$, Volume weighted mean temperament of the Arctic oce Global surface mean DIC, Mean surface chlorophyll in the |
| *Seven Levels of Climate Change* 2YE9uHBE5OI | 14-10-2019 | 2:55 | PI control, historical, SSP1 1.9, SSP1 2.6, SSP4 3.4, SSP5 3.4 - overshoot, SSP2 4.5, SSP3 7.0, SSP5 8.5 | Global mean SST, pH, Drake Passage current, Global mean surface chlorophyll, Global total Air sea flux of $CO_2$, Global total Ice extent |

**Table 2.** Table showing the number of unique viewers, total views, the average view duration, the audience retention and the total time spent watching each video. The data in this table covers the 90 day range from the 21st of August to the 18th of November 2019

| Video | Unique Viewers | Total Views | Average View Duration (minutes:seconds) | Retention % | Watch Time (hours) |
|---|---|---|---|---|---|
| Earth System Allegro | 143 | 213 | 0:38 | 61 | 2.3 |
| Pre-industrial Vivace | 97 | 136 | 0:50 | 34 | 1.9 |
| Ocean Acidification in E minor | 68 | 89 | 0:59 | 51 | 1.5 |
| Sea Surface Temperature Aria | 37 | 59 | 0:52 | 68 | 0.9 |
| Giant Steps Spin Up | 21 | 31 | 1:39 | 58 | 0.9 |
| Seven Levels of Climate Change | 19 | 25 | 1:29 | 51 | 0.6 |
| Total | 251 | 553 | 0.51 | 53 | 8.0 |