# Peer review of "Earth System Music: the methodology and reach of music generated from the United Kingdom Earth System Model (UKESM1)"

_Geoscience Communication, 2019_

## Short Comment (SC1) · 19 Jan 2020

In my years following scientific research, I have no idea what the meaning of this is https://www.youtube.com/watch?v=RxBhLNPH8ls

Music appreciation is subjective, but scientific research results should not be. Sorry if I don't get what the point is.

---

## Referee Comment (RC1) · Anonymous Referee #1 · 17 Feb 2020

Comments on: "Earth system Music: the methodology and reach of music generated from the United Kingdom Earth System Model".

The article is well written and very interesting. Below we provide feedback to the author's that we hope is useful. This was co-reviewed by myself and a colleague, and we ourselves are a science-art collaborative team. This co-review of the article was agreed by Editor Sam Illingworth.

1. As this is a pilot study, it is clear why only one tool was utilised to gather and analyse data towards the reach, engagement and audience of the channel and the videos. Nevertheless, it would be very useful for readers and those who might like to expand

on this methodology and methods if the choice to not triangulate the data was stated as one of the limitations, and discussed. You could highlight in the introduction that this is a pilot study, stating it explicitly, and then follow-up with a brief discussion about the experimental character of the research and why in this instance the focus was not on triangulating findings, but to present the study. If you are planning additional evaluation and analyses it would be useful to highlight that, even if briefly, in the discussion.

2. The authors' choices for tempo, genre, and scale of each composition (allegro, vivace, aria etc.) adds to the "emotional connection" mentioned in line 213. There is value in the authors elaborating on these choices, and to explain if the reason of each choice is due to the feeling that the data are expressing. For example, for Earth System Allegro, one can notice that the authors describe it as "[. . .] a future scenario in which the anthropogenic impact on the climate is at a minimum" (line 225). This could be perceived as a happy scenario, and is potentially why the authors aligned it with the allegro rhythm, because allegros are usually lively and merry tempos, able to express and communicate positive and happy scenarios. It would add richness to the methods and explanation of data interpretation for the reader if more details like those that I expressed above could be included in the descriptions of all the pieces.

3. More specifically, in the Quantification of Reach section more detail could be provided to support the evaluation component of this work. We recognize this is a pilot study, but even then there is the potential to include perspectives shared by others (qualitative data), and to use these data as a starting point to build a stronger understanding of how this work is 'reaching' others. For example, you didn't include data on shares or people's comments or shared perceptions (qualitative data) about the project as shared through particular social media platforms. While I agree it can be useful to have demographic data, it wasn't clear from the start of the paper that you were interested in the demographic of people that this work reaches. Equally, you report on the nation that YouTube viewers were from in your results, which is fairly limited demographic data, and don't include other data despite having noted its availability in your

methods section. From a reader's perspective it isn't only interesting how many people the work reached or who those people are, but also what their perspectives of the work was and any messages that emerged from viewers that could inform our broader understanding about what people took from viewing and experiencing this work. It would help readers to know more about the 'experience' in addition to the 'reach' and the authors could begin to form this with a content analysis of the comments or perspectives shared, and even brief quotes of feedback and perspectives shared by others to offer some insight to people's perspectives.

4. We include a few more specific questions about the quantification of reach below. p2 line 52 "provides additional contextual clues to aid the interpretation" Please elaborate on why this happens e.g. the animated graphs provide information usually not available or not attractive enough to read for the general public. p7 line 159 "The conversion from model data to musical pitch is performed in two stages. First [...]" Please clarify what is the second stage. p7 line 154 "[...] is an artistic choice" Is there a concept behind that artistic choice each time? Does something trigger the composer to choose a specific scale and not choose another? It help the reader to know if there are any creative or conceptual reasons behind these choices or if it is mostly due to the desirable harmony or aesthetics of the final composition. p14 line 355 How was the playlist shared? P14 line 341 Note that the authors highlight that most views occurred after the first few days, and this is also presented again in the results. It would be better to highlight this as either an element of the method or as a result, but not as both. p14 line 355-57 Please provide reasoning for why broader networks were not engaged and why press releases weren't used to disseminate the research or playlists. This is a primary stream of sharing research, and media teams are trained in helping to guide how research and outputs are shared and delivered to others. Could your reach have been greater had you employed / utilized the resources available through those existing networks? Equally, paying to transmit the research and outputs would likely increase the number of people the work 'reaches', why was this avoided? If the goal was to get the research to different people then using those tools would assist, and it is then not too surprising

that the research was primarily viewed only in the first few days; this could potentially have been overcome by having invested more into the transmission of the research and working with different traditional and social media tools. More explanation is needed as to why different tools weren't used, and potentially for the authors to consider if the approach used to evaluate 'reach' might be better removed from this particular paper that is heavily focused on the methodology and method, and included in a subsequent text that explores evaluation, reach, and viewer experience in greater detail and in a more robust way. p18 line 420 "[. . .] Sea surface temperature aria" Consistency note: The titles are presented in italic and with the initial letter capital throughout the article. Please change to: Sea Surface Temperature Aria.

---

## Referee Comment (RC2) · Anonymous Referee #2 · 5 Mar 2020

Comments on:"Earth System Music: the methodology and reach of music generated from the United Kingdom Earth System Model by Mora et al.

This manuscript describes six music pieces that have been produced to make climate data accessible to non-experts. The aims are (1) to generate music pieces using climate model data, (2) to use music to illustrate standard practices in Earth System modelling to non-experts, and (3) to quantify the dissemination of music pieces. The method employed here (i.e., data sonification or turning data into music) is a powerful approach and have been successfully used by others for presenting complex datasets to engage/inspire those outside the expert community. The method is particularly well-

suited for working with climate data. The authors have done a thorough job of explaining each music video. However, there are several sections that need improvements (particularly the method and evaluation). Therefore, I see two major changes:

1. Of the above three goals (also stated in the manuscript), goal 1 was clearly achieved. However, it is not clear how this manuscript addresses goal 2. As for goal 3, there was no systematic, robust documentation of the authors' dissemination strategy, the audience demographics, learning, etc. Therefore, a more quantitative (and systematic) assessment of the video usage is needed. The authors state some strategies for doing this (in discussion and conclusion, e.g., performing to a live audience and surveying the audience to measure impact). I think such strategies are great and should be implemented. It is difficult to indicate if a science communication product is helpful (and if so in what way) without any systematic assessment. Therefore, I highly encourage the authors to consider evaluating their videos, and adding the analysis of their findings to the manuscript.

2. The method section (lines 106-192) is difficult to follow for non-musicians. This section explains how the music was produced but fails to explain how it relates to climate data. I think giving some examples may increase its readability. For example, when you state "the lowest value in the dataset is presented by the lowest note..." (line 120), it may be helpful to give an example of the lowest value in the dataset (e.g., coldest recorded temperature). The same goes for the highest value in the dataset (line 121). Line 119 also says "each model dataset is linked to a series of notes", so does this mean each note is a data point? Again, translating this into climate data would be helpful. Similar suggestions for Fig 2 (see below) and lines 169-172. Also, many of the comments shown in the next few pages are related to this issue.

Figure 2- It would be very helpful if you can connect what you show in the piano keyboard to climate data. See figure 1 of George et al. (2017, American Meteorological Sociey) for example. Again what does each note represent? What does each pitch represent? A bit hard to follow as a non-musician.

Below, my comments are shown line by line:

Line 23 (Introduction)-The authors introduce the topic well, and the references they list are relevant and helpful. Since this study combines sonification with imagery, it would also be helpful to know if this approach has been taken before, and if so, how does this study contribute (or build on) previous work?

Line 56 points out the potential for biased-interpretation of data using sonification. However, the authors do not return to this issue later to discuss it. Was this a concern during this study and how was it addressed?

Figure 1. Though flow charts are generally produced in this way, I suggest to add a few images (one per section) to draw in the readers. The sections are: datasets (top), music (middle) and videos (bottom).

Line 224. When you state "The Earth System Allegro is a relatively fast-paced piece in C Major", can you describe what C Major sounds like for non-musicians? Also the rest of the sentence starting with "...showing some important metrics of the happy to keep..." does not make grammatical sense. Please revise.

Line 226. Could you explain how this video demonstrates the principles of sonification using the data series?

Line 232. I think there may be a typo here. Could it be "year 2030/2040" as oppose to year 2100?

Line 235. Consider deleting this sentence as it is repetitive.

Line 240. Consider deleting the sentence starting with "Effectively, ..." It is redundant.

Line 248. Change "there's" to "there is", and "doesn't" to "does not" in line 292. And reflect this change throughout the manuscript.

Line 250. Again "a very common 4 chord song: C Major, G Major, A Minor, and F Major" does not mean anything to a non-musician. Please clarify this by giving an
example for each or give a word to describe what they sound like.

Lines 250-253. Draft a similar paragraph for section 3.1.1. This helps connect the music structure with what the dataset represents.

Line 256. Add a reference to Figure 3, pane 3, at the end of this sentence.

Lines 257-259. Add the name of scenarios (e.g., SSP5 8.5) to Figure 3, pane 3.

Line 270. Add reference to Figure 3, pane 4, after "E minor".

Line 273. How are these 15 historical simulations are shown in the figure? Only 6 lines are shown. Have they been grouped?

Line 274. "This piece uses a repeating 12 bar blues structure in E minor", what does this mean to a non-musician, and how is this connected to the dataset it is reflecting?

Line 285. What initial conditions are the authors referring to?

Line 286. When you state "...the results of our projections are due to changes..." what changes are the authors referring to?

Line 294. Please give an example of what it is meant by "inherent change" and "underlying drift".

Line 295. The spin up ran for 5000 simulated years. Why 5000 years? How was this time selected? A reference is provided, but it would be useful to add a sentence explaining why.

Line 305. It may be useful to label these lines in Figure 3, pane 5 or describe them in the caption.

Line 311. Why was the musical progression slowed to one chord per four beats? What does it mean in terms of the climate dataset?

Line 335. Decapitalize "Global total ice" and insert a space between "extent" and "blue".
Line 336. Change "view" to "video" in "the percentage of the view that the average audience viewed".

Line 366-369. Consider deleting this part starting from "Aside from the metrics…" These numbers are too small to be meaningful, and are not discussed.

Lines 370-378. Consider deleting the whole paragraph or move to discussion.

Figure 7. Consider removing it from the manuscript, but keep the text (lines 387 onward). The figure does not add much to the manuscript, especially when half of the data is unknown.

Lines 390-392. Consider deleting this or move to discussion.

Line 406. The study goals stated here differ from those stated in page 10. Please keep the goals consistent.

Line 411. The authors conclude that once the concept was demonstrated, there was reduced enthusiasm from the audience to return to it. How do they know that? Another possibility could be that the audience didn't feel the need to return to it, or it could also be that the videos sparked their interest further so that they ended up checking out similar videos outside the playlist. These are all possibilities, and there is no evidence for or against them. I suggest sticking to the facts, and only interpret the data when it is actually possible (which is not the case here).

Line 412. The last sentence may be true but it is irrelevant to this paragraph. Was the goal to grow a YouTube channel? Why do the authors mention this here?

Lines 420-426. The sea surface temperature aria is also the most visually simple animation when compared with the rest. The viewer is not required to keep track of multiple datasets and listen to the music at the same time. Could this be also why this piece has the highest audience retention?

Line 430. There is no documented evidence that the music pieces and animations

improved the wider public's understanding of climate change modelling. The authors mention this in the next paragraph. So I suggest to delete the "perhaps, improve the wider public's understanding of climate change modelling". One could hope for that, but this study was not designed to assess that, and certainly did not do that.

I suggest to move some of the content currently placed in the discussion section to two new sections: Limitations and Future Work. This means most of what is shown in page 19-20 can be reorganized to fit into one of these two sections. This might help the readers. Line 439. Here the authors suggest hosting live events to fully explain the methodology used by the modelling community. But is this something non-experts are interested in, or is this the aim of this study? I thought the idea was to use a unique communication method (sonification and imagery) to explain complex datasets to non-experts. If this method requires a live event for further explanation, then it does not fulfill what it was supposed to do: to engage non-experts.

Line 462. The authors state that it was hard to distinguish the different datasets in the music. One solution would be to insert a very short silence in music between different datasets. Just an idea.

Line 504. Insert a space between the word viral and the references.

Line 510. Insert a space between the word afternoons and the references.

Line 514. "they reached an audience of 251 unique viewers and a total view count of 553"

Table 1. Add unit of time for "duration". Minutes?

---

## Author Comment (AC1) · 17 Apr 2020

Dr. Lee de Mora
Plymouth Marine Laboratory
Prospect Place
The Hoe
Plymouth
PL1 3DH

Dear Geoscientific Communication editors, referees and reviewers,

We received two review comments and one short comment. We have addressed their comments in this document, and made changes to the main document, also attached. We're like to thank both anonymous referees and the short comment author for their helpful comments. Thanks to their contributions, this work is in a much better state and should be easier to follow.

For clarity, we will reproduce the comments and respond to them in turn. Our responses are marked in bold and begin with **"LdM:".** The new text are included with an indent and may include some latex grammar. Alternatively, a document showing the difference between the old and the new version is also available.

Sincerely,

Lee de Mora – representing the authorship team.

**Anonymous Referee #1**

The article is well written and very interesting. Below we provide feedback to the author's that we hope is useful. This was co-reviewed by myself and a colleague, and we ourselves are a science-art collaborative team. This co-review of the article was agreed by Editor Sam Illingworth.

**LdM: Thank you both for your review and for the kind words. Also, thanks to Sam for allowing this novel team review approach.**

1. As this is a pilot study, it is clear why only one tool was utilised to gather and analyse data towards the reach, engagement and audience of the channel and the videos. Nevertheless, it would be very useful for readers and those who might like to expand on this methodology and methods if the choice to not triangulate the data was stated as one of the limitations, and discussed. You could highlight in the introduction that this is a pilot study, stating it explicitly, and then follow-up with a brief discussion about the experimental character of the research and why in this instance the focus was not on triangulating findings, but to present the study. If you are planning additional evaluation and analyses it would be useful to highlight that, even if briefly, in the discussion.

**LdM: At the request of referee #2, we added a limitations section. That goes into more detail about this. We've also added a short paragraph to the introduction to explicitly point out that this is a pilot study.**

> It should be noted that this work is an early pilot study. The aims of the project are outlined below in sect.~\ref{sec:works}. The limitations of this approach are outlined in sect.~\ref{sec:limitations}.

2. The authors' choices for tempo, genre, and scale of each composition (allegro, vivace, aria etc.) adds to the "emotional connection" mentioned in line 213. There is value in the authors elaborating on these choices, and to explain if the reason of each choice is due to the feeling that the data are expressing. For example, for Earth System Allegro, one can notice that the authors describe it as "[. . .] a future scenario in which the anthropogenic impact on the climate is at a minimum" (line 225). This could be perceived as a happy scenario, and is potentially why the authors aligned it with the allegro rhythm, because allegros are usually lively and merry tempos, able to express and communicate positive and happy scenarios. It would add richness to the methods and explanation of data interpretation for the reader if more details like those that I expressed above could be included in the descriptions of all the pieces.

**LdM: We have added additional explanations of how the artistic choices were made for each piece in their descriptions. These are:**

[revised manuscript text omitted]

3. More specifically, in the Quantification of Reach section more detail could be provided to support the evaluation component of this work. We recognize this is a pilot study, but even then there is the potential to include perspectives shared by others (qualitative data), and to use these data as a starting point to build a stronger understanding of how this work is 'reaching' others. For example, you didn't include data on shares or people's comments or shared perceptions (qualitative data) about the project as shared through particular social media platforms.

**LdM: We have added a section to the results that list all the comments on social media.**

The following statement were posted via social media. These were posted directly on the YouTube video page, or on social media posts linking to the video via facebook, twitter and reddit. Note that we have removed emojis and gifs, but otherwise reproduced comments as they originally appeared.

\begin{enumerate}

\item This one was very dramatic.

\item It gets quite dramatic after 1950-60

\item This was submitted as a scientific research paper here, but don't understand what the point is.

\item Wow! That's awesome!

\item Great idea

\item That's Crazy!!

\item Awesome! Personally I was hoping to hear something of the same ilk as system of a down or similar. Does make modelling sound far more upbeat though

\item Brilliant idea! Just like Herman Hess's book 'The Glass bead Game', where data is unified from many sources and brought together by many senses. Listen to how the ocean sings!

\item AMAZING!!!!

\item A total new meaning for the "listen to the ocean" motto

\item That's awesome! Can you post this to someone with a bigass twitter handle. This deserves more attention. Maybe send it to Adam Rutherford at Inside Science?

\item It's quite different from the previous one! Super Cool stuff!!

\item This is amazing. If you can find a setup that would cause noticeable change in the music between pre-industrial and future you're viral

\item I love this!! Well done!

\item This is amazing ! Was initially quite surprised that these weren't more tonally chaotic - then I read I the blurb for the top video. I presume that you could do the same with any choice of key and scale? Also, I hope you realise that, with your choice of C-major, you have made something worthy on inclusion in the next Axis of Awesome medley

\item Certainly very interesting, but it lacks a lot of human touch. These generations don't seem to take into account many compositional techniques that are almost vital to make a comprehensive piece of music, such as motifs, dynamics, musical form, things like that. I found that there wasn't much for the ear to catch onto musically, no clear melodies, rhythms, harmonic progressions etc. Because of this it can also be quite hard for a human to learn, adding on to the challenge of coordinating the strange rhythms between the hands. So it's cool, but very unhuman music and thus most likely difficult for humans to play.

\item I think if humans would take their time to learn this, the musical phrasing they would bring to the table could do a lot to make the pieces easier and more enjoyable to listen to. I think computer made music is very cool in that it really highlights the amount of complexity that goes into creating music as we would know it, namely through the human mind. So the kind of research you're conducting is very valuable indeed!

\item Wow this actually sounds very interesting! Sadly I'm not a very accomplished pianist either, but I'll leave this comment and upvote for visibility.

\end{enumerate}}

**And I've expanded the discussion section about the audience**

> The comments from social media were listed above in sect.~\ref{sec:results} and were almost all positive and supportive. However, these comments are biased towards the authors friends, family and professional colleagues. These comments include several positive comments and praise, comments about the pieces themselves (``it gets dramatic at 1950-1960``), comparisons to other musicians or works of art (e.g. System of a down, Herman Hess's book Glass bead game, the Axis of Awesome medley ), one person ``didn't understand what the point is``, and the final three comments were from a music forum and contain some interesting insight into the musical side of the work. While we hoped to disseminate information about Earth System modelling to a wider audience, it's not possible to determine whether the audience learned anything about Earth System modelling using the metrics provided by YouTube studio or the comments posted on social media. Furthermore, it is not possible to determine whether the audience was composed of laymen or experts. As this was a pilot study, we did not go into greater detail to understand the audience reactions. Future extensions of this project should include a survey of the audience, investigating their backgrounds, demographics, what they learned about Earth System models and their overall impressions of the pieces. This could take the form of an online survey associated with each video, or a discussion with the audience at a live performance event.

While I agree it can be useful to have demographic data, it wasn't clear from the start of the paper that you were interested in the demographic of people that this work reaches.

**LdM: I would include the demographics of the audience under the wider term, "reach", which is appears in the abstract and even in the title. Nevertheless, I added the word demographics to the following sentence in the introduction:**

> This toolkit allows content creators to monitor the reach, engagement and audience demographics (age, gender, country of origin) for their channel as a whole, as well as for individual videos.

Equally, you report on the nation that YouTube viewers were from in your results, which is fairly limited demographic data, and don't include other data despite having noted its availability in your methods section. From a reader's perspective it isn't only interesting how many people the work reached or who those people are, but also what their perspectives of the work was and any messages that emerged from viewers that could inform our broader understanding about what people took from viewing and experiencing this work. It would help readers to know more about the 'experience' in addition to the 'reach' and the authors could begin to form this with a content analysis of the comments or perspectives shared, and even brief quotes of feedback and perspectives shared by others to offer some insight to people's perspectives.

**LdM: We have added the section above and an discussion on the viewer feedback.**

4. We include a few more specific questions about the quantification of reach below.

p2 line 52 "provides additional contextual clues to aid the interpretation" Please elaborate on why this happens e.g. the animated graphs provide information usually not available or not attractive enough to read for the general public.

**LdM: This paragraph confused two points and was split up to read:**

> With the ever-growing interest from the general public towards understanding climate science, it is becoming increasingly important that we present this information in ways accessible to non-experts. It is also becomingly increasingly easier for scientists to use tools such as social media to engage with non-experts audiences and the wider public.

**And the section about contextual clues to aid the interpretation was removed.**

p7 line 159 "The conversion from model data to musical pitch is performed in two stages. First [. . .]" Please clarify what is the second stage.

**LdM: The text here was changed to:**

> The conversion from model data to musical pitch is performed in using the following method.

p7 line 164 "[. . .] is an artistic choice" Is there a concept behind that artistic choice each time? Does something trigger the composer to choose a specific scale and not choose another? It help the reader to know if there are any creative or conceptual reasons behind these choices or if it is mostly due to the desirable harmony or aesthetics of the final composition.

**LdM: We have expanded following section to the methods section to more accurately describe the role of the artist in creating the piece:**

> While the method is relatively straightforward and repeatable, each piece has a diverse range of settings and artistic choices made by the composer: the choice of datasets used to determine pitch and velocity for each track, the pitch and velocity ranges for each track, the piece's tempo and the number of notes per beat, the musical key and chord progression for each track, and the width of the smoothing window. The choice of instrument is also another artistic choice, although in this work, only one instrument was used, the TiMidity+ piano synthesizer. As a whole, these decisions allow the composer to attempt to define the emotional context of the final piece. For instance, a fast-paced piece in a major progression may sound happy and cheerful to an audience who are used to associating fast-paced songs in major keys with happy and cheerful environments. It should be mentioned that there are no strict rules governing the emotional context of chords, tempo or instrument and the emotional contexts of harmonies, timbres and tempos differ between cultures. In a scientific context, exploiting the western musical traditions can allow the composer to imbue the piece with the associated emotional musical cues.

p14 line 355 How was the playlist shared?

**LdM: The text was changed to:**

> Each piece was also added to a YouTube playlist, which was shared via social media in the same way as the individual pieces.

P14 line 341 Note that the authors highlight that most views occurred after the first few days, and this is also presented again in the results. It would be better to highlight this as either an element of the method or as a result, but not as both.

**LdM:  This fits more in the results section than in the methods section, so it was removed from the methods section.**

p14 line 355-57 Please provide reasoning for why broader networks were not engaged and why press releases weren't used to disseminate the research or playlists. This is a primary stream of sharing research, and media teams are trained in helping to guide how research and outputs are shared and delivered to others. Could your reach have been greater had you employed / utilized the resources available through those existing networks? Equally, paying to transmit the research and outputs would likely increase the number of people the work 'reaches', why was this avoided?

**LdM: In practice, this was a zero-budget trial study. We were not able to budget the staff time in the communications team, nor were we able to purchase reach through youtube advertising.  We have raised these are potential future avenues to reach a wider audience.**

**The text was changed to:**

> As this was a pilot study, there were no particular timed-releases, no press release and no direct assistance from the PML, NERC or UKESM communications teams. The videos were then disseminated through these networks and allowed to reach wider audiences. In addition, no paid advertisements were purchased. Please see sect. 6 for a discussion on possible extensions to this pilot study where these avenues could be explored.

**Also note that we plan to produce a press release should this paper be accepted for final publication. That pulibcation and subsequent press release may become a new startring point to count views for this work. We may also time the publication of a new piece to coincide.**

If the goal was to get the research to different people then using those tools would assist, and it is then not too surprising that the research was primarily viewed only in the first few days; this could potentially have been overcome by having invested more into the transmission of the research and working with different traditional and social media tools. More explanation is needed as to why different tools weren't used, and potentially for the authors to consider if the approach used to evaluate 'reach' might be better removed from this particular paper that is heavily focused on the methodology and method, and included in a subsequent text that explores evaluation, reach, and viewer experience in greater detail and in a more robust way.

**LdM:  We have changed to goals to more accurately reflect how we view these works. This work was more of an exploration of the use of music to communicate science rather than an outreach experiment with pre-defined research questions. We learned about it as we went and explored**

**what was possible. In this work, we hope to explain some of the things that worked and some that didn't for us. It should be noted that none of the authors have experience in outreach research.**

p18 line 420 "[. . .] Sea surface temperature aria" Consistency note: The titles are presented in italic and with the initial letter capital throughout the article. Please change to: Sea Surface Temperature Aria.

**LdM: Fixed.**

---

## Author Comment (AC2) · 17 Apr 2020

Dr. Lee de Mora
Plymouth Marine Laboratory
Prospect Place
The Hoe
Plymouth
PL1 3DH

Dear Geoscientific Communication editors, referees and reviewers,

We received two review comments and one short comment. We have addressed their comments in this document, and made changes to the main document, also attached. We're like to thank both anonymous referees and the short comment author for their helpful comments. Thanks to their contributions, this work is in a much better state and should be easier to follow.

For clarity, we will reproduce the comments and respond to them in turn. Our responses are marked in bold and begin with **"LdM:".** The new text are included with an indent and may include some latex grammar. Alternatively, a document showing the difference between the old and the new version is also available.

Sincerely,

Lee de Mora – representing the authorship team.

**Anonymous Referee #2**

This manuscript describes six music pieces that have been produced to make climate data accessible to non-experts. The aims are (1) to generate music pieces using climate model data, (2) to use music to illustrate standard practices in Earth System modelling to non-experts, and (3) to quantify the dissemination of music pieces. The method employed here (i.e., data sonification or turning data into music) is a powerful approach and have been successfully used by others for presenting complex datasets to engage/inspire those outside the expert community. The method is particularly well suited for working with climate data. The authors have done a thorough job of explaining each music video.

**LdM: Thank you for the summary and the kind words.**

However, there are several sections that need improvements (particularly the method and evaluation). Therefore, I see two major changes:

1. Of the above three goals (also stated in the manuscript), goal 1 was clearly achieved. However, it is not clear how this manuscript addresses goal 2. As for goal 3, there was no systematic, robust documentation of the authors' dissemination strategy, the audience demographics, learning, etc. Therefore, a more quantitative (and systematic) assessment of the video usage is needed. The authors state some strategies for doing this (in discussion and conclusion, e.g., performing to a live audience and surveying the audience to measure impact). I think such strategies are great and should be implemented. It is difficult to indicate if a science communication product is helpful (and if so in what way) without any systematic assessment. Therefore, I highly encourage the authors to consider evaluating their videos, and adding the analysis of their findings to the manuscript.

**LdM: We have added more detail to the quantification of reach, results, discussions and limitations sections. There are  many changes throughout the paper and we refer the editor and reviewer to the difference document, which lists these changes.**

2. The method section (lines 106-192) is difficult to follow for non-musicians. This section explains how the music was produced but fails to explain how it relates to climate data. I think giving some examples may increase its readability. For example, when you state "the lowest value in the dataset is presented by the lowest note. . ." (line 120), it may be helpful to give an example of the lowest value in the dataset (e.g., coldest recorded temperature). The same goes for the highest value in the dataset (line 121).

**LdM: We have expanded this whole section and made many clarifications. Unfortunately, music theory is its own discipline (as is music composition) and it's not possible to make a complete, thorough, brief and easy to follow explanation here.**

Line 119 also says "each model dataset is linked to a series of notes", so does this mean each note is a data point? Again, translating this into climate data would be helpful. Similar suggestions for Fig 2 (see below) and lines 169-172. Also, many of the comments shown in the next few pages are related to this issue.

**LdM: In addition to adding the new keyboard figure (see below), I've changed this entire paragraph to:**

> Each model timeseries dataset is converted into a series of consecutive MIDI notes, which form a track. For instance, the Sea Surface Temperature (SST) time series could be converted into a series of MIDI notes in the upper range of the keyboard, forming a track. For each track, the time series data is converted into musical notes such that the lowest value in the dataset is represented by the lowest note pitch available, and the highest value of the dataset is represented by the highest pitch note available. The notes in between are assigned proportionally by their data value between the highest and lowest pitched notes. The lowest and highest notes available for each track are pre-defined in the piece's settings. Each track is given its own customised pitch range, so that the tracks may be lower pitch, higher pitch or have overlapping pitch ranges relative to other tracks in the piece. The ranges of notes available for the piece \textit{Earth System Allegro} is shown in fig.~\ref{fig:histograms}. In this figure, the four histograms on the left hand side show the distributions of data used in the piece, and the right hand side shows a standard piano keyboard which the musical range available to each dataset. For instance, the Drake Passage Current ranges between 135 and 175 Tg s$^{-1}$ in these simulations and we selected a range between MIDI pitches 72 and 96. This means that the lowest Drake passage current values (135 Tg s$^{-1}$) would be represented in MIDI with a pitch of 72 and the highest Drake passage current values (175 Tg s$^{-1}$) would be assigned a MIDI pitch of 96, which is two octaves higher.

Figure 2- It would be very helpful if you can connect what you show in the piano keyboard to climate data. See figure 1 of George et al. (2017, American Meteorological Sociey) for example. Again what does each note represent? What does each pitch represent? A bit hard to follow as a non-musician.

**LdM: Based on the 2017 paper in the Bulletin of the American Meteorological Society by St George et al (**_https://doi.org/10.1175/BAMS-D-15-00223.1_**) I've added the following image and caption to this paper:**

[Figure]

**Caption:**

The musical range of each of the datasets used in the Earth System Allegro. The four histograms on the left hand side show the distributions of data used in the piece, and the right hand side shows a standard piano keyboard which the musical range available to each dataset. In this piece, the Drake passage current, shown in red, is free to vary within a two octave range of the C major scale. The other three datasets have their own ranges, but are limited to the notes in the chord progression C major, G major, A minor F major. The dark coloured keys are the notes in C major, but the lighter coloured keys show the other notes which are available. Note that both the C major scale and chord do not include any of the ebony keys on a piano, but these notes would be used if they are within the available range.

Below, my comments are shown line by line:

Line 23 (Introduction)-The authors introduce the topic well, and the references they list are relevant and helpful. Since this study combines sonification with imagery, it would also be helpful to know if this approach has been taken before, and if so, how does this study contribute (or build on) previous work?

**LdM: Added the following text.**

It should be noted that all the pieces list here are also accompanied by a video which can explain the methodology behind the creation of the music, shows the performance by the artists, or shows the data development while the music is played.

Line 56 points out the potential for biased-interpretation of data using sonification. However, the authors do not return to this issue later to discuss it. Was this a concern during this study and how was it addressed?

**LdM: Upon reflection, we never wanted the musical pieces to be a neutral objective version of the data. We always wanted to try to communicate some of the emotional context of the data.**

**Re-wrote this paragraph to be:**

> In addition to its practical applications, sonification is a unique field where scientific and artistic purposes may coexist \citep{Tsuchiya2015}. This is especially true when in addition to being converted into sound, the data is also converted into music. This branch of sonification is called musification. Through the choice of musical scales and chords, tempo, timbre and volume dynamics, the composer adds emotive meaning to the piece. As such, unlike sonification, musification should be treated as a potentially biased-interpretation of the underlying data. It can not be a true objective representation of the data. Note that the philosophical distinction between sound and music is beyond the scope of this work. Furthermore, even though the composer may have made musical and artistic decisions to link the behaviour of the data with an emotive state, it may not necessarily be interpreted in the same way by the listener.

Figure 1. Though flow charts are generally produced in this way, I suggest to add a few images (one per section) to draw in the readers. The sections are: datasets (top), music (middle) and videos (bottom).

**LdM: We have reworked the methods plot now. It should be both clearer, more colourful and include a few images.**

[Figure]

**The new caption reads:**

> The computational process used to convert UKESM1 data into a musical piece and associated video. The boxes with a dark border represent files and datasets, and the arrows and chevrons represent processes. The blue areas are UKESM1 data and the pre-processes stages, the green areas show the data and processing stages needed to convert model data into MIDI data, and orange area show the post processes stages which convert images and MIDI into sheet music and video.

Line 224. When you state "The Earth System Allegro is a relatively fast-paced piece in C Major", can you describe what C Major sounds like for non-musicians? Also the rest of the sentence starting with ". . .showing some important metrics of the happy to keep. . ." does not make grammatical sense. Please revise.

**LdM: Changed this paragraph to:**

> The Earth System Allegro is a relatively fast-paced piece in C Major, showing some important metrics of the Southern Ocean in the recent past and projected into the future with the SSP1 1.9. This is the future scenario in which the anthropogenic impact on the climate is at a minimum. The C major scale is composed of only natural notes (no sharp or flat notes), making it one of the first chords that people encounter when learning music. In addition, major chords and scales like C Major typically sound happy. Christian Schubart's `Ideen zu

einer Aesthetik der Tonkunst` (1806) describe C major as "Completely pure. Its character is: innocence, simplicity, naivety, children's talk." As this was the first piece in the series, the link between this seemed an appropriate way to start the Earth System Music project. Through choosing C major and an upbeat tempo, and data from the best possible climate scenario (SSP1 1.9), we aimed to start the project with a piece with a sense of optimism about the future climate and to introduce the principles of musification of UKESM1 time series data.

Line 226. Could you explain how this video demonstrates the principles of sonification using the data series?

**LdM: Changed this phrase to:**

introduce the principles of musification of UKESM1 time series data.

**This was the first piece in the series and does introduce the core-concept of the project, that the music follows the data.**

Line 232. I think there may be a typo here. Could it be "year 2030/2040" as oppose to year 2100?

**LdM:  Changed this phrase to:**

Even under SSP1 1.9, UKESM1 predicts that this value would rise from around zero during the pre-industrial period to maximum of approximately 2 Pg of carbon per year around the year 2030, followed by a return to zero at the end of the century.

Line 235. Consider deleting this sentence as it is repetitive.

**LdM: removed and replaced with:**

The fourth field is the Southern Ocean mean surface temperature, shown in green, which slightly rises from approximately 5 degrees in the pre-industrial period up to a maximum of 6 degrees.

Line 240. Consider deleting the sentence starting with "Effectively, . . ." It is redundant.

**LdM:  Removed.**

Line 248. Change "there's" to "there is", and "doesn't" to "does not" in line 292. And reflect this change throughout the manuscript.

**LdM:  Done**

Line 250. Again "a very common 4 chord song: C Major, G Major, A Minor, and F Major" does not mean anything to a non-musician. Please clarify this by giving an example for each or give a word to describe what they sound like.

**LdM: Added the sentence which should add some context.**

> This chord progression is strikingly popular and may be hear in songs such as: Let it Be, by the Beatles, No Woman no Cry by Bob Marley and the Whalers, With or without you by U2, I'm yours by Jazon Mraz, amoung many others.

**This is a bit of a trick, as some of these songs are the same chord progression in a different key (For instance, Africa by Toto is written in A Major). I fear that adding the complexity of the roman numeral notation is a step too far for this work! Out of interest, the following songs are written using this progression:**
https://en.wikipedia.org/wiki/List_of_songs_containing_the_I%E2%80%93V%E2%80%93vi%E2%80%93IV_progression

Lines 250-253. Draft a similar paragraph for section 3.1.1. This helps connect the music structure with what the dataset represents.

**LdM: Done, see above.**

Line 256. Add a reference to Figure 3, pane 3, at the end of this sentence.

**LdM: Done**

Lines 257-259. Add the name of scenarios (e.g., SSP5 8.5) to Figure 3, pane 3.

**LdM: I'm not able to do this, these figures show the final frame of each video, not the data itself. As the video is already published, it's not possible to do this. The colour code is described in the text in lines 255-260.**

Line 270. Add reference to Figure 3, pane 4, after "E minor".

**LdM: added the sentence:**

> The final frame of this video is shown in pane 4 of fig. 3.

Line 273. How are these 15 historical simulations are shown in the figure? Only 6 lines are shown. Have they been grouped?

**LdM: As mentioned above, figure 3 only shows the final frame of the videos. I've changed figure 3's caption to be clearer:**

> Figure 3: The final frame of each of the six videos. These frames of the videos are shown in the order that they were published. The videos 1), 3), 5) and 6) use a consistent x-axis for the

duration of the video, but videos 2) and 4) have x-axes which changes over the course of the video. This means that panes 2 and 4 show only a small part of time range.

Line 274. "This piece uses a repeating 12 bar blues structure in E minor", what does this mean to a non-musician, and how is this connected to the dataset it is reflecting?

**LdM: Added the following paragraph to make the point clear about the choice of a 12 bar blues.**

> This piece uses a repeating 12 bar blues structure in E minor and a relatively fast tempo. This chord progression is was exceptionally common progression, especially in the blues, Jazz and early rock n roll sounds. It is composed of four bars of the E minor, two bars of A minor, 2 bars of E minor, then one bar of B minor, A minor, E minor and B minor. The twelve bar blues can be be heard in songs such as: Johnny B. Goode by Chuck Berry ,Hound Dog by Elvis Presley, I got you (I feel Good) by James Brown, Sweet Home Chicago by Robert Johnson or Rock n Roll by Led Zeppelin. In the context of Earth System Music, the 12 bar pattern with its opening set of four bars, then two sets of two bar and ending for four sets of one bar between key changes drives the song forward before starting again slowly. This behaviour is thematically similar to the behaviour of the ocean acidification in UKESM1 historical simulation, where the bulk of the acidification occurs at the end of each historical period.

Line 285. What initial conditions are the authors referring to?

**LdM: Changed the text to**

> When we produce models of the Earth System, we use a range of points of the pre-industrial control as the initial conditions for the historical simulations. All the historical simulations have slightly different starting points, and evolve from these different initial conditions, which gives us more confidence that the results of our projections are due to changes since the pre-industrial period instead of simply a consequence of the initial conditions.

Line 286. When you state ". . .the results of our projections are due to changes. . ." what changes are the authors referring to?

**LdM: see above**

Line 294. Please give an example of what it is meant by "inherent change" and "underlying drift".

**LdM: I've re-written this paragraph for greater clarity and this should address most of the issues raised in this section.**

> This piece combines the spin up of the United Kingdom Earth System Model with the chord progression of John Coltrane's Giant Steps. The spin up is the process of running the model from a set of initial condition to an equilibrium steady state. When a model reaches a steady state, this means that there is no significant trend or drift in the mean behaviour of several key metrics. For instance, as part of the C4MIP protocol, Jones et al (2016) suggest a drift criterion of less than 10 Pg of Carbon per century in the absolute value of the flux of $CO_2$

from the atmosphere to the ocean. In practical terms, the ocean model is considered to be spun up when the long-term average of the air sea flux of Carbon is consistently between -0.1 and 0.1 Pg of carbon per year.

The spin up is a crucial part of model development. Without spinning up, the historical ocean model would still be equilibrating with the atmosphere. It would be much more difficult to separate the trends in the historical and future scenarios from the underlying trend of a model still trying to equilibrate. Note that while a steady state model does not have any significant long term trend or drifts; it can still have short term variability. This short term variability can be seen in the pre-industrial simulation in the Pre-industrial Vivace piece. It can take a model thousands of years of simulation for the ocean to reach a steady state. In our case, the spin up ran for approximately 5000 simulated years before the spun up drift criterion were met , Yool 2020.

Line 295. The spin up ran for 5000 simulated years. Why 5000 years? How was this time selected? A reference is provided, but it would be useful to add a sentence explaining why.

**LdM: I've added more details on the spin up criteria, please see above. Also note that the UKESM spin up paper has recently been submitted to JAMES after review and is expected to be accepted for final publication before this paper. A suitable reference will be added when it is available.**

Line 305. It may be useful to label these lines in Figure 3, pane 5 or describe them in the caption.

**LdM: These lines do appear in figure 3, pane 5, I've added the following text to the caption.**

Pane 5 includes two vertical lines showing the jumps in the spin up piece. Pane 6 shows a single vertical line for the crossover between the historical and future scenarios.

Line 311. Why was the musical progression slowed to one chord per four beats? What does it mean in terms of the climate dataset?

**LdM: To be honest, this was a happy accident due to a bug in the original code. The original version at full speed just sounded too chaotic. Changed the text to:**

This change occurred as an accident, but we found that the full speed version sounded very chaotic, so the slowed version was published instead.

Line 335. Decapitalize "Global total ice" and insert a space between "extent" and "blue".

**LdM: Done**

Line 365. Change "view" to "video" in "the percentage of the view that the average audience viewed".

**LdM: Done**

Line 366-369. Consider deleting this part starting from "Aside from the metrics. . ." These numbers are too small to be meaningful, and are not discussed.

**LdM: The first reviewer had a stronger interest in metrics beyond the youtube statistics. We have moved this part into that section and extended the discussion to include these metrics.**

Lines 370-378. Consider deleting the whole paragraph or move to discussion.

**LdM: Moved to discussion.**

Figure 7. Consider removing it from the manuscript, but keep the text (lines 387 onward). The figure does not add much to the manuscript, especially when half of the data is unknown.

**LdM: Figure removed but kept the text.**

Lines 390-392. Consider deleting this or move to discussion.

**LdM: Split this paragraph into two and put second half in the discussion.**

Line 406. The study goals stated here differ from those stated in page 10. Please keep the goals consistent.

**LdM: Fixed.**

Line 411. The authors conclude that once the concept was demonstrated, there was reduced enthusiasm from the audience to return to it. How do they know that? Another possibility could be that the audience didn't feel the need to return to it, or it could also be that the videos sparked their interest further so that they ended up checking out similar videos outside the playlist. These are all possibilities, and there is no evidence for or against them. I suggest sticking to the facts, and only interpret the data when it is actually possible (which is not the case here).

**LdM: Removed this sentence.**

Line 412. The last sentence may be true but it is irrelevant to this paragraph. Was the goal to grow a YouTube channel? Why do the authors mention this here?

**LdM: Removed this sentence**

Lines 420-426. The sea surface temperature aria is also the most visually simple animation when compared with the rest. The viewer is not required to keep track of multiple datasets and listen to the music at the same time. Could this be also why this piece has the highest audience retention?

**LdM: added:**

> The sea surface temperature aria is also the most visually simple animation of the six pieces. Only one pane is visible in the video and much of the piece only includes one or two voices at a time. It may be possible that this simplicity holds the audience's attention.

Line 430. There is no documented evidence that the music pieces and animations improved the wider public's understanding of climate change modelling. The authors mention this in the next paragraph. So I suggest to delete the "perhaps, improve the wider public's understanding of climate change modelling". One could hope for that, but this study was not designed to assess that, and certainly did not do that.

**LdM: Changed text to:**

> While we hoped to improve the wider public's understanding of the methods used in climate change modelling, the tools available to us within YouTube studio do not allow any way of assessing this. Please see the Limitations and Future Work section, below.

I suggest to move some of the content currently placed in the discussion section to two new sections: Limitations and Future Work. This means most of what is shown in page 19-20 can be reorganized to fit into one of these two sections. This might help the readers.

**LdM: Created this section and reorganised the discussion into two parts.**

Line 439. Here the authors suggest hosting live events to fully explain the methodology used by the modelling community. But is this something non-experts are interested in, or is this the aim of this study? I thought the idea was to use a unique communication method (sonification and imagery) to explain complex datasets to nonexperts. If this method requires a live event for further explanation, then it does not fulfill what it was supposed to do: to engage non-experts.

**LdM: At the moment, the videos by themselves do not even attempt to include an explanation of the methods used in Earth System Modelling. All explanation was in the video description below the video. However, the point of this paragraph is confused by the addition of a sentence about a live performance, so it was removed.**

**We changed this paragraph to:**

> The videos themselves only include the music and a visualisation of the data, they do not include any description about how the music was generated or the Earth system modelling methods used to create the underlying data. The explanations of the science and musification methodologies are held in a text description below the video. Furthermore, viewers must expand this box by clicking the ``show more`` button. Using YouTube studio, it is not currently possible to determine whether the viewers have expanded, read or understood the description. When we have shown these videos to live audiences at scientific meetings and conferences, it has always been associated with a brief explanation of the methods. In the future, this explanatory preface to the work could be included in the video itself or as a separate video.

Line 462. The authors state that it was hard to distinguish the different datasets in the music. One solution would be to insert a very short silence in music between different datasets. Just an idea.

**LdM: The confusion here is that one sentence is trying to cover two ideas. We changed this sentence to:**

> These pieces were all performed by the same instrument, a solo piano, which limits the musical diversity of the set of pieces. In addition, each dataset within in a given piece was performed by the same instrument, making it difficult to distinguish the different datasets being performed simultaneously.

Line 504. Insert a space between the word viral and the references.

**LdM: done**

Line 510. Insert a space between the word afternoons and the references.

**LdM: done**

Line 514. "they reached an audience of 251 unique viewers and a total view count of 553"

**LdM: done**

Table 1. Add unit of time for "duration". Minutes?

**LdM:  done**

---

## Author Comment (AC3) · 17 Apr 2020

Dr. Lee de Mora
Plymouth Marine Laboratory
Prospect Place
The Hoe
Plymouth
PL1 3DH

Dear Geoscientific Communication editors, referees and reviewers,

We received two review comments and one short comment. We have addressed their comments in this document, and made changes to the main document, also attached. We're like to thank both anonymous referees and the short comment author for their helpful comments. Thanks to their contributions, this work is in a much better state and should be easier to follow.

For clarity, we will reproduce the comments and respond to them in turn. Our responses are marked in bold and begin with **"LdM:".** The new text are included with an indent and may include some latex grammar. Alternatively, a document showing the difference between the old and the new version is also available.

Sincerely,

Lee de Mora – representing the authorship team.

**Short Comment: Paul Pukite**

**We have also received the following comment from Paul Pukite:**

> In my years following scientific research, I have no idea what the meaning of this is https://www.youtube.com/watch?v=RxBhLNPH8ls Music appreciation is subjective, but scientific research results should not be. Sorry if I don't get what the point is.

**Paul has also added a comment on the youtube channel, and we have been communicating with him directly there. We wrote:**

> The idea here is to use music to draw people in, then they learn about how Earth System Modelling works. This piece was the first one in a series of six, and the main idea here is just to show how the musification process works. Ie, we take Earth System Model data and turn it into music.

> The next piece in the series introduces the concept of a control run. Then the following piece shows how future scenarios work in the context of global warming, then there's a piece about ocean acidification and how historical runs branch from the pre-industrial control run. There's also a piece about how the model is spun up, and another about the 7 SSP scenarios in CMIP6. These are all key concepts in climate modelling, but might not known outside our community. The goal isn't really to use the data to identify new behaviours in the model, but to show how we make models in a (hopefully) fun way.

> Happy to answer more questions if you have any!

---

## Referee Report (RR1)

Earth System Music: the methodology and reach of music generated from the United Kingdom Earth System Model (UKESM1) –
By de Mora et al. (2020)

I reviewed and posted my referee comment in the interactive discussion of the earlier version of this manuscript on March 5, 2020. The authors have addressed most of my comments, and the revised version is more concise and readable. I particularly like the edits done to fig 1 and fig 2 as well as the text revisions in section 3.1 (Works) which make the content more accessible to non-musicians. My comments related to the current version of this manuscript are shown below. I have two main comments; comment #1 requires major revisions (and is preferred):

1. **Evaluation Strategy (Major revision) -** This study has three main goals: (1) to turn data into music, (2) to show standard practices in climate modelling, and (3) to quantify the dissemination of musical pieces. The manuscript addresses the first two goals well, but (as stated in my earlier comments) it does not include a systematic and robust analysis that is needed for tackling goal 3. The collection and analysis of data from YouTube's channel's monitoring toolkit (the way it is done in the manuscript) are not enough to effectively quantify their reach, even for a pilot study like this. To me, this evaluation strategy appears to be an afterthought, and can be significantly improved to make the results more interesting and meaningful to the readers.

Therefore, I suggest one of the followings:

> (A) Focus the manuscript on goal 1 and 2. Skip goal 3. This would make this manuscript a method-focused paper.

> (B) Include goal 3. This will require additional work (1-3 months). I realize that the current pandemic imposes limits on how much work can be done at the moment. However, using online surveys/interviews could still be a very good option to collect dissemination data. One idea would be for the authors to start collecting data on video usage from Earth sciences students and/or Earth sciences community including EGU/AGU members. Survey questions could focus on whether or not each video meets its objectives. These objectives should be clearly defined in the paper.

> My impression is that much of the underlying practices in climate modelling is unfamiliar even to Earth scientists / students – therefore, if the videos cannot reach Earth scientists unfamiliar with climate modelling, it would be very hard to reach the general public. Therefore, testing these videos with the Earth sciences community could be a very good step in that direction.

2. **Paper Reorganization** - The 6 musical pieces and videos are currently described in the methods section. I suggest moving section 3.1 (and all the subsections: 3.1.1 - 3.1.6) to the results section because the videos are the final products (i.e. results) of the methods described in section 2. Then remove what is currently in the results section (i.e. evaluation data which is not the strength of this paper) to the discussion section under a new subheading (e.g., video assessment).

Below, I have listed some minor edits:

**Minor edits:**

1. Line 142: change "which" to "with"

2. Throughout manuscript, change "data is" to "data are"

3. Perhaps consider combining figure 2 and 3.

4. Consider deleting the two logos from figure 1. They're unnecessary.

5. Delete line 210 – unnecessary sentence: "While the method is relatively straightforward and repeatable" and start the paragraph with "Each piece has a diverse …"

6. Delete lines 258-259 – unnecessary sentence: "As this was the first piece in the series, this seemed an appropriate way to start the Earth System Music project."

7. Sometimes the authors use "fig" or "figure" in the text to refer to figures. Please pick one and keep it consistent throughout the manuscript.

8. References section – it would be great if the actual YouTube links can be added either in the manuscript or in the reference section so the readers can look them up. For examples, the following two references contain no links:

Line 705: de Mora, L.: Lee de Mora's YouTube channel homepage, 2019.
Line 738: Reddit: Data is beautiful sub-reddit., 2019.

9. Line 254: delete the extra "is" in "The SSP1 1.9 projection is  the future scenario in…"

10. Line 271 – I think a verb is missing from this sentence: "In this piece, the Drake Passage current is set to the C major scale, but the other three parts module between the C major, G major, A minor and F major chords." Or are you using "module" as a verb?

11. Line 298 - "The first scenario..." instead of "The three scenarios"

12. Define SSP somewhere in the text.

13. Line 299: delete the extra "apostrophe" in "business as usual"'

14. Line 325 - remove the extra "be"

15. Line 334: I think you mean "ocean surface" as opposed to "surface ocean"

16. Line 339 - Change "our" to "scientific" in "outside our community"

17. Line 358: Change "criterion were" to "criterion was"

18. Line 388: delete the second "and"

19. Line 406 – There are some minor grammatical errors in this sentence: "YouTube's built-in toolkit for channel monitoring, YouTube studio, was used "to" quantify the reach, "audience" engagement for the individual videos, as well as the entire channel (Google, 2019)."

20. In addition to comment 19, I suggest to delete "audience engagement" all together unless you consider clicking and maybe viewing a video a meaningful form of engagement. This study mainly assesses the number of viewers as opposed to evaluating audience engagement.

21. Line 407: "Using these tools…" I think you mean "using this tool…."

22. Lines 408-409: In regards to this sentence: "This is because YouTube studio can only produce reliable data on unique viewers for a period up to 90 days." Any idea why? Readers would be interested in this. Please elaborate.

23. Line 410: If a person watches the same video from the same device multiple times, is this counted only as one view or multiple views. Not clear from the sentence.

24. Line 448-454: This entire paragraph reads like a figure caption/legend. I suggest deleting it and adding it to the figure caption.

25. Line 503: "they" not "the" in "…they included a wide range of…"

26. Out of curiosity – are there good references for evaluating YouTube videos using the YouTube monitoring toolkit? Can you reference them? For example, in line 557, the authors state "We have found that the YouTube studio toolkit is not currently sufficiently capable to fully quantify the reach of these videos." Are there other studies making the same claim? And if so, what kind of changes are needed to improve the toolkit?

27. I highly recommend remove "social media statements" from the manuscript main text as they don't add much to the manuscript. Alternatively, include them as supplementary material. Same goes for paragraph 546-551.

28. Paragraph 525 - Change the start sentence to "The total number of views of the Earth System music playlist is comparable with other videos on Earth System 525 modelling."

29. Line 561 – The authors state that "it is not possible to determine whether the audience learned anything about Earth System modelling using the metrics provided by YouTube studio or the comments posted on social media." – This shows that the evaluation strategy was not successful in assessing "learning" and/or "engagement". But this should have been clear from early stages of this study. Again, the evaluation section of this paper is weak, and can be significantly improved using other methods.

30. Line 563 – Just because the work is a pilot study, it doesn't mean that one should not assess its effectiveness robustly especially if the assessment is set as a goal of the study. I suggest revising this sentence.

31. Line 592 – Revise sentence "we were not able to vary  the instruments used"

32. Line 594 – "In addition, each dataset  in a given piece was…"

33. Line 549 – "…it has been advised…" not "it's been advised"

34. The caption for Table 1 appears to be wrong.

35. In the supplementary materials (zip file), two music sheets appear to be missing.

---

## Author Response (AR2)

GC-2019-28 - second major revision
**Authors Response**

Thanks to the GC editors and the anonymous referees for all their hard work reviewing this manuscript.

After careful consideration, we have decided to follow the recommendation of the editor, Sam Illingworth. His advice was to reformat the paper and follow up with a secondary paper at a later date with a more detailed evaluation of the dissemination and its impact.

We have also implemented all the minor corrections requested by the anonymous Reviewers in this second round of major revisions.

Thanks,

Lee de Mora and the authors of GC-2019-28.

[revised manuscript text omitted]